# Disruption of the autoinhibited state primes the E3 ligase parkin for activation and catalysis

Atul Kumar[1],[†], Jacob D Aguirre[2],[†], Tara EC Condos[2],[†], R Julio Martinez-Torres[1],[†], Viduth K Chaugule[1], Rachel Toth[1], Ramasubramanian Sundaramoorthy[3], Pascal Mercier[2], Axel Knebel[1], Donald E Spratt[2], Kathryn R Barber[2], Gary S Shaw[2],[*] & Helen Walden[1],[**]

## Abstract

The *PARK2* gene is mutated in 50% of autosomal recessive juvenile parkinsonism (ARJP) cases. It encodes parkin, an E3 ubiquitin ligase of the RBR family. Parkin exists in an autoinhibited state that is activated by phosphorylation of its N-terminal ubiquitin-like (Ubl) domain and binding of phosphoubiquitin. We describe the 1.8 Å crystal structure of human parkin in its fully inhibited state and identify the key interfaces to maintain parkin inhibition. We identify the phosphoubiquitin-binding interface, provide a model for the phosphoubiquitin–parkin complex and show how phosphorylation of the Ubl domain primes parkin for optimal phosphoubiquitin binding. Furthermore, we demonstrate that the addition of phosphoubiquitin leads to displacement of the Ubl domain through loss of structure, unveiling a ubiquitin-binding site used by the E2~Ub conjugate, thus leading to active parkin. We find the role of the Ubl domain is to prevent parkin activity in the absence of the phosphorylation signals, and propose a model for parkin inhibition, optimization for phosphoubiquitin recruitment, release of inhibition by the Ubl domain and engagement with an E2~Ub conjugate. Taken together, this model provides a mechanistic framework for activating parkin.

**Keywords** enzyme mechanism; Parkinson's disease; phosphorylation; ubiquitination; ubiquitin ligase
**Subject Categories** Molecular Biology of Disease; Post-translational Modifications, Proteolysis & Proteomics; Structural Biology
The EMBO Journal (2015) 34: 2506–2521

See also: **KK Dove** *et al* (October 2015)

## Introduction

Parkinson's disease (PD) is a neurodegenerative disorder with severe motor and non-motor symptoms (Goedert *et al*, 2013). There are several genes associated with PD, with autosomal recessive juvenile parkinsonism (ARJP) being the most prevalent familial form of the disease. Mutations in *PARK2* and *PARK6* genes lead to ARJP, with *PARK2* mutations causing > 50% of cases (Bonifati *et al*, 2002). *PARK2* encodes parkin, an E3 ubiquitin ligase, while *PARK6* encodes PTEN-induced kinase 1 (PINK1), which phosphorylates Ser65 of parkin and ubiquitin.

Parkin is an RBR-type ubiquitin ligase which is characterized by *R*ING1, in*B*etweenRING (IBR) and *R*ING2 domains. Recent studies have suggested these names do not accurately reflect the structures or functionality of the domains. As a result, an alternative nomenclature that uses *R*ING1, *B*enign Rcat (BRcat=IBR) and *R*equired for catalysis (Rcat=RING2) domains has been proposed for all RBR E3 ligases (Spratt *et al*, 2014) that maintain the familiar RBR scheme. RBR ligases contain domains additional to the signature RBR module that are specific to each member. Parkin has an N-terminal ubiquitin-like domain (Ubl) that shares 65% sequence similarity with ubiquitin, and a zinc-chelating RING0 domain immediately N-terminal to the RBR domains (Hristova *et al*, 2009). The RING1 domain recruits E2 enzymes, which carry charged ubiquitin and pass it to a catalytic cysteine in the parkin RING2 domain (Wenzel *et al*, 2011). Parkin can function with multiple E2s and catalyses the formation of several modifications including K63-linked and K48-linked chains, and monoubiquitination. Pathogenic mutations occur throughout Parkin, and mutations in different domains have different effects on activity, solubility and stability (Wang *et al*, 2005; Hampe *et al*, 2006; Safadi & Shaw, 2007; Safadi *et al*, 2011; Spratt *et al*, 2013). Wild-type parkin exists in an autoinhibited state (Chaugule *et al*, 2011); subsequently, autoinhibition has been demonstrated to be common to RBR ligases (Smit *et al*, 2012; Stieglitz *et al*, 2012; Duda *et al*, 2013).

1   MRC Protein Phosphorylation and Ubiquitylation Unit, College of Life Sciences, University of Dundee, Dundee, UK
2   Department of Biochemistry, Schulich School of Medicine & Dentistry, University of Western Ontario, London, ON, Canada
3   Centre for Gene Regulation and Expression, College of Life Sciences, University of Dundee, Dundee, UK
    *Corresponding author. Tel: +1 519 661 4021; E-mail: gshaw1@uwo.ca
    **Corresponding author. Tel: +44 1384 384109; E-mail: h.walden@dundee.ac.uk
    [†]These authors contributed equally to this work
    [The copyright line of this article has been changed on 14 October 2015 after original online publication.]

The N-terminal Ubl domain inhibits parkin activity, and pathogenic mutations in this domain render parkin active (Chaugule et al, 2011). High-resolution crystal structures of parkin's RING0-RBR domains, which lack the Ubl domain, revealed multiple domain–domain interactions that regulate parkin activity (Chaugule et al, 2011; Riley et al, 2013; Trempe et al, 2013; Wauer & Komander, 2013). However, the importance of the Ubl domain is highlighted by recent findings that PINK1, the kinase encoded by *PARK6*, phosphorylates parkin at serine 65, and this phosphorylation enhances parkin activity (Kondapalli et al, 2012; Shiba-Fukushima et al, 2012, 2014). PINK1 also modifies ubiquitin at the equivalent serine 65 (Kane et al, 2014; Kazlauskaite et al, 2014b; Koyano et al, 2014), and this signal is necessary for parkin activation. Together, these phosphorylation signals enable parkin to ubiquitinate multiple substrates, including several involved in mitochondrial homoeostasis.

Despite the recent breakthroughs in our understanding of parkin structure and activation, we still do not have a clear picture of how the Ubl domain maintains the tertiary inhibited conformation of wild-type parkin and how this inhibition is disrupted by phosphorylation of both the Ubl domain and ubiquitin to activate parkin. Since many disease-causing mutations are localized to the Ubl domain, understanding how the Ubl controls the inhibition of parkin at the molecular level will likely be the key to developing activator therapeutic molecules that may be useful in treating the disease. In this work, we set out to understand how the Ubl domain regulates parkin function and how the phosphorylation signals impact parkin activity. We report here the 1.8 Å resolution crystal structure of human parkin, complete with the Ubl domain. Our structural and biochemical analysis reveals that the interface between the Ubl and RING1 domains is crucial for parkin's regulation by remotely altering the arrangement of the RING0/RING1 interface, giving rise to a compact conformation. Furthermore, we determine the 2.3 Å structure of active phosphomimetic (S65D) parkin, which reveals a conformation closer to that of the truncated parkin structures. Combining comprehensive thermodynamic, biophysical and structural analyses, we find that the role of the Ubl domain is to prevent parkin function in the absence of the phosphoubiquitin signal. We show that phosphorylation of the Ubl domain optimizes parkin for phosphoubiquitin binding at the interface between the RING0 and RING1 domains and present an NMR-derived model of the parkin–phosphoubiquitin complex. Further, we identify that phosphoubiquitin binding leads to displacement of the Ubl domain, revealing a ubiquitin recognition site used to assist recruitment of the ubiquitin–E2 complex. These findings establish a framework for understanding the cycle of parkin inhibition and activation, provide a molecular rationale for the observed E2 promiscuity and present a scaffold for targeting domain–domain interactions to activate parkin.

# Results

Although there are crystal structures of truncated parkin comprising the RING0, RING1, IBR and RING2 domains (R0RBR parkin), and a 6.5 Å low-resolution structure of full-length parkin (Riley et al, 2013; Trempe et al, 2013; Wauer & Komander, 2013), there are no high-resolution molecular details of parkin with the Ubl domain

intact. Given the importance of this domain, and observations that ARJP mutations within this domain lead to increased ubiquitination activity (Chaugule et al, 2011), we aimed to identify the molecular determinants of the autoinhibition. To re-affirm the autoinhibitory role of the Ubl domain and to test the importance of the linker, we conducted ubiquitination assays using an improved fluorescence-based method (Fig 1A). These experiments repeatedly showed that removal of the Ubl domain (77-C, 111-C) leads to increased parkin autoubiquitination as well as formation of free polyubiquitin chains that was enhanced upon stimulation with phosphoubiquitin. In contrast, more complete removal of the linker (127-C, 141-C (R0RBR)) appears to reverse this process leading to similar ubiquitination levels as the wild-type protein, perhaps indicating that the N-terminus of the linker may play an undiscovered role in ubiquitination by parkin.

Attempts to crystallize the full-length protein did not generate high-quality crystals, so we undertook a rational approach to provide details of the Ubl domain interaction with the rest of the protein. Previous work has shown that the linker region between the Ubl and RING0 domains (residues 77–141) in parkin is likely poorly structured with little known function (Hristova et al, 2009). In addition, the linker is the most divergent region for parkin when sequences are compared from multiple species (Fig EV1). This suggested that careful deletion of the linker might improve crystallization without eliminating the inhibitory functionality of the Ubl domain. We generated a series of human parkin proteins with modified linker lengths (Fig 1B) between the Ubl domain and RING0-RBR domain to test their ubiquitination activity and ability to form high-quality crystals. In particular, one construct (UblR0RBR) that included the Ubl (residues 1–83) linked to RING0RBR (residues 144–465) showed autoinhibition similar to the wild-type protein, enhanced ubiquitination with phosphoubiquitin and the ability to ubiquitinate a potential mitochondrial substrate protein, Miro1 (Kazlauskaite et al, 2014a) (Fig 1B).

## The structure of parkin reveals a compact, autoinhibited conformation

The crystal structure of UblR0RBR parkin was solved and refined to 1.8 Å, and comprises two molecules in the asymmetric unit (Table 1). The structure has five domains: Ubl (1–76), RING0 (144–216), RING1 (228–327), IBR (328–377) and RING2 (415–465), and a repressor element (REP) (Fig 1C). Residues 354 and 357–360 in the IBR domain are missing in the density, as are the residues 383–390 linking the IBR to the REP, and residues 406–413 linking the REP to the RING2 domain.

The position of the Ubl domain, located between the RING1 and IBR domains, is globally consistent with the low-resolution 6.5 Å structure of full-length rat parkin (Trempe et al, 2013) (Appendix Fig S1). The largest interface in the structure is that formed between the Ubl domain and the rest of parkin, burying ~2,150 Å$^2$ accessible surface area (Fig 2A). The primary contact is between the Ubl (β3, β5) and RING1 (helix H1) domains (Fig 2A; Fig EV1). The core interface is stabilized by hydrophobic interactions mediated by I44 and V70 of the Ubl domain and L266, V269 and T270 of RING1 (Fig 2A). This interaction is supported by several key salt bridges and hydrogen bonds including R42 (Ubl) with D262 at the

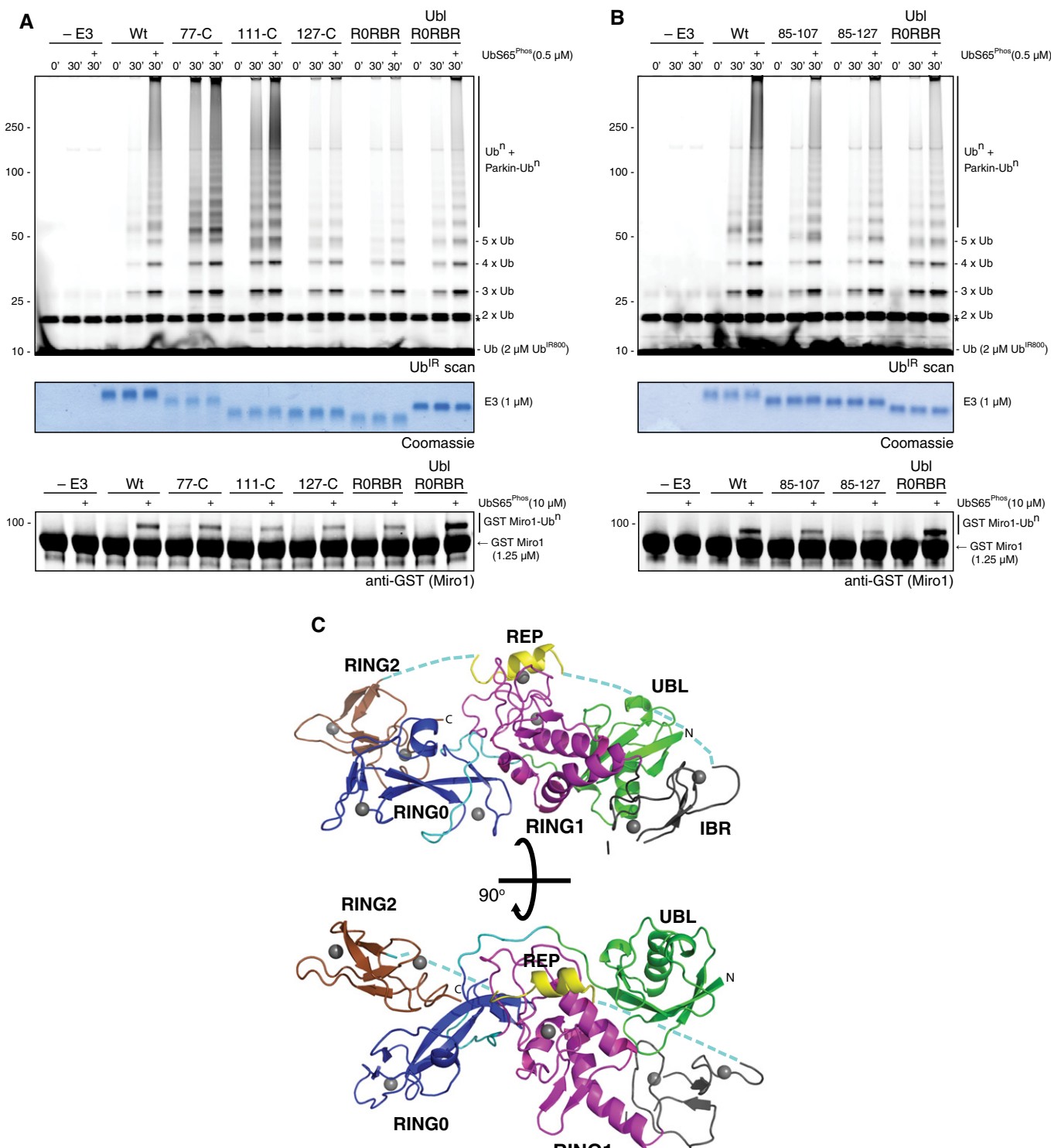

**Figure 1.  Three-dimensional structure of UblR0RBR parkin.**

A   Ubiquitination assays show wild-type and UblR0RBR parkin are inhibited for chain formation (top) and Miro1 ubiquitination (bottom) in the absence of phosphoubiquitin at zero time point and in the presence of phosphoubiquitin at 30 min. Removal of the inhibitory Ubl domain leads to parkin activation dependent on the length of the linker preceding RING0. The middle panel shows a Coomassie-stained loading control. A non-specific, ATP-independent band is indicated (*).

B   Ubiquitination assays show effects of varying length of the linker between the inhibitory Ubl domain and R0RBR, in the presence and absence of phosphoubiquitin. A non-specific, ATP-independent band is indicated (*).

C   Overall structure of UblR0RBR parkin depicting the Ubl domain (green), RING0 (blue), RING1 (magenta), IBR (black), REP (yellow) and RING2 (brown). Loops outside domains are in cyan. Zinc atoms are represented as grey spheres. Two views are shown related by a 90° rotation in *x*.

**Table 1.  Crystallographic data collection and refinement statistics.**

| | UblR0RBR | S65DUblR0RBR |
|---|---|---|
| Wavelength (Å) | 0.9174 | 0.9172 |
| Resolution (Å) | 56.36–1.79 (1.85–1.79) | 23.16–2.37 (2.46–2.37) |
| Space group | P2₁2₁2₁ | P2₁2₁2₁ |
| Unit cell abc (Å) αβγ (°) | 67.3 67.3 206.2 90 90 90 | 66.3 66.3 205.6 90 90 90 |
| Total reflections | 276,440 | 216,582 |
| Unique reflections | 87,891 | 37,844 |
| Multiplicity | 3.1 (2.8) | 5.7 (5.0) |
| Completeness (%) | 98.8 (97.6) | 99.8 (98.8) |
| Mean I/sigma(I) | 11.2 (1.8) | 7.7 (1.5) |
| Wilson B-factor (Å²) | 30.7 | 46.0 |
| R-merge | 4.9 (56.5) | 17.4 (116.9) |
| R-meas | 6.7 (77.0) | 19.1 (130.6) |
| CC1/2 | 99.6 (75.3) | 99.3 (55.4) |
| Reflections for R-free | 4,270 | 1,854 |
| R-work | 0.192 (0.224) | 0.220 (0.241) |
| R-free | 0.212 (0.227) | 0.241 (0.267) |
| CC (work) | 0.951 | 0.912 |
| CC (free) | 0.946 | 0.905 |
| Number of non-hydrogen atoms | 6,943 | 6,424 |
| Macromolecules | 6,180 | 6,055 |
| Ligands | 52 | 56 |
| Water | 711 | 313 |
| Protein residues | 784 | 784 |
| RMS bonds Å | 0.008 | 0.007 |
| RMS angles (°) | 0.94 | 0.87 |
| Ramachandran favoured (%) | 96.4 | 95.0 |
| Average B-factor (Å²) | 49.1 | 52.9 |
| Macromolecules | 48.2 | 53.0 |
| Ligands | 44.1 | 62.0 |
| Water | 57.3 | 49.6 |

Numbers in parentheses indicate outer shell statistics.

N-terminus of helix H1 (RING1) and R6 and H68 (Ubl) with N273 and D274 at the C-terminus of helix H1. The tail of the Ubl domain is stabilized by interactions between the main chain carbonyl oxygens of R75 and K76 with R245 of RING1. In addition, N8 (Ubl) of the β1–β2 loop sits between E310 and Q311 at the extreme N-terminus of the bent helix (H3) connecting the RING1 and IBR domains (Fig 2A). There are few contacts between the Ubl and IBR domains, the most obvious involving H11 (β1–β2 loop, Ubl) with P333 and K369 (IBR). The intimate Ubl/RING1 association also occurs in *trans*, based on solution NMR experiments where the Ubl domain was titrated into R0RBR parkin (residues 141–465). The largest chemical shift changes in R0RBR occur in residues in helix H1, and helix H3 in the RING1 domain (Fig 2B). In addition,

chemical shift changes occur for residues A383, S384 and G385 in the tether region following the IBR domain (Fig 2B) that is not observed in the crystal structure, indicating these residues interact with the Ubl domain transiently. Reverse titration experiments that monitored changes in ${}^{1}$H-${}^{13}$C HMQC spectra of the Ubl domain upon R0RBR addition show the methyl groups of residues I2, I44, A46, L61, I66 and V70 have the largest changes, consistent with I44 and V70 facing the RING1 domain as in the crystal structure (Appendix Fig S2) and I2, A46, L61 and I66 facing the tether. Taken together, these data indicate an extensive interface between the Ubl domain and the RING1 domain of parkin that exists both in *cis* and in *trans*, in solution and in the crystal.

The extent and nature of the Ubl/RING1 interface suggests it is important for autoinhibition of parkin. Consistent with this, several activating ARJP mutations occur in the Ubl domain including R42P, A46P and R33Q (Chaugule *et al*, 2011). The R42P mutation would disrupt the central salt bridge between R42 and D262 (Fig 2A). The ARJP mutation of R33Q would abolish a long-range ionic interaction with E370 in the IBR domain. Finally, mutation of A46P would induce a loss of secondary structure in the loop that interacts with the tether region (Appendix Fig S2). Weakening or destabilizing of these interactions that maintain the Ubl position would lead to a loosening of its tight interface with RING1. To test the importance of this interface, we generated several Ubl domain mutants and assayed them for activity (Fig 2C). In contrast to wild-type Parkin, Ubl-interface mutants are more active for chain formation and ubiquitination of Miro1 (Kazlauskaite *et al*, 2014a). Taken together, these data show that disruption of the Ubl-RING1 interface leads to constitutively active parkin.

## Presence of the Ubl domain maintains a compact RING0–RING1 interface

Several groups reported high-resolution crystal structures of parkin R0RBR domains from human and rat, lacking the Ubl domain (Riley *et al*, 2013; Trempe *et al*, 2013; Wauer & Komander, 2013). In order to understand what conformational changes, if any, occur beyond the Ubl/RING1 interface when the Ubl is present, we compared UblR0RBR parkin with the truncated R0RBR structures. The structures agree well (Appendix Fig S3) with root mean squared differences (rmsd) between 1.4 Å (rat R0RBR, 4K7D) and 1.8 Å (human R0RBR, 4I1F). The observed differences are due largely to the positions of the IBR domains in each structure (Appendix Fig S3). Heteronuclear nOe experiments show this domain exhibits the greatest degree of flexibility of any domain in R0RBR parkin (Appendix Fig S4). The REP is located in the equivalent position, and the RING0/RING2 interface is maintained between structures (Appendix Fig S3). Interestingly, when the structures are superimposed using the RING0/RING2 domains as the reference point, the flexibility of the other domains is more pronounced (Fig 3A). In particular, the RING0/RING1 interface widens in the absence of the Ubl domain as the IBR domain swings ~12 Å away from its position when the Ubl domain is attached. These global changes suggest a pivot or hinge in parkin centred near the RING0/RING1 interface. In UblR0RBR, the RING0/RING1 interface buries 730 Å² and involves Y147, Y149, K151, Q155, V157, W183 and L187 of the RING0 domain interacting with H227, I229, A230, P247, E300, H302, R305,

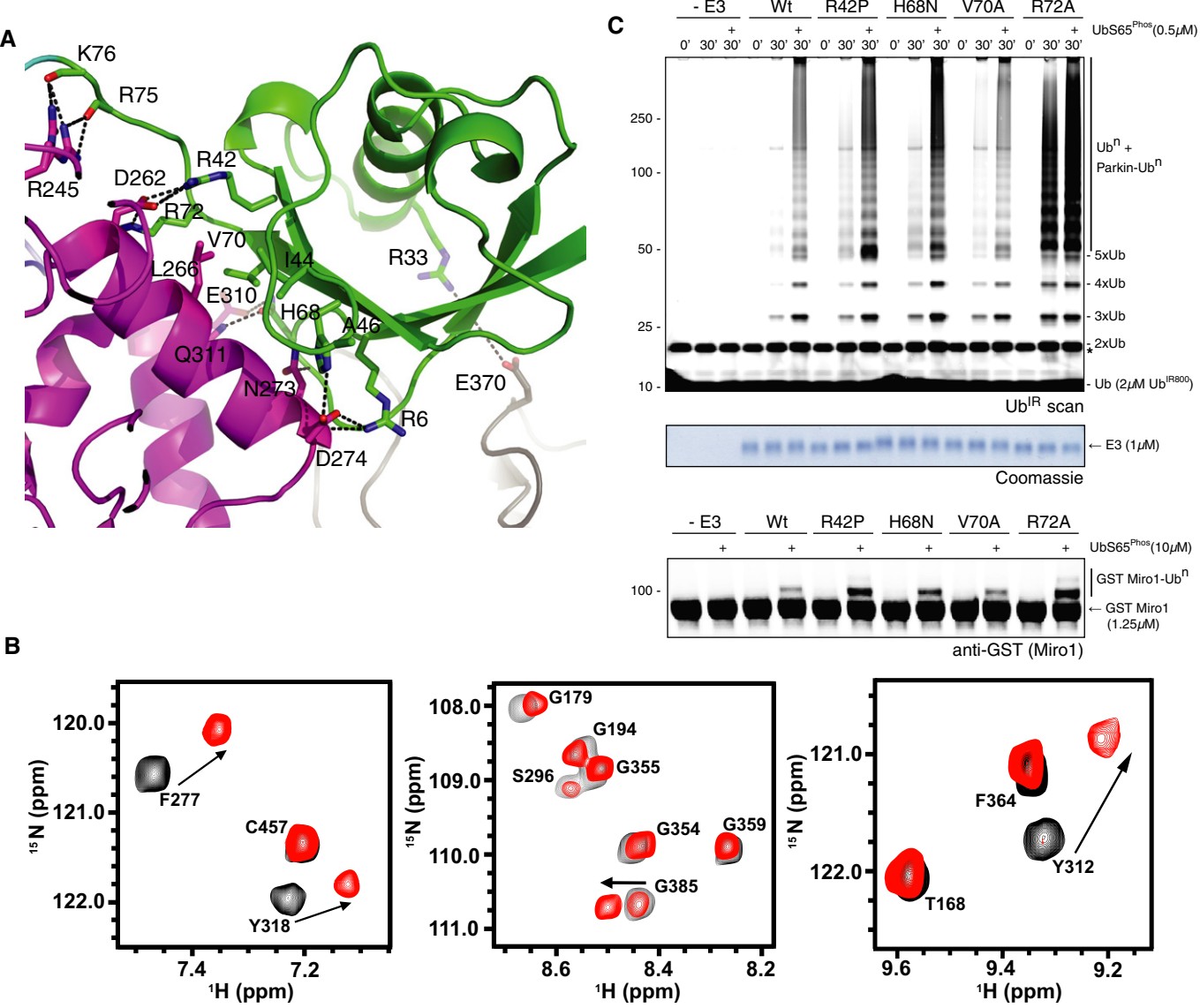

**Figure 2. The molecular determinants of Ubl-mediated autoinhibition.**

A Close-up of the Ubl/RING1 interface in parkin UblR0RBR in the same colours and orientation as Fig 1C bottom. Hydrogen bonds and salt bridges in the interface are shown.

B Selected regions from 600 MHz $^1$H-$^{15}$N TROSY spectra showing 200 μM $^2$H,$^{13}$C,$^{15}$N-labelled R0RBR (black contours) following the addition of 300 μM of the Ubl domain (red contours). Residues shown that undergo chemical shift changes are F277 (RING1, helix H1), Y312 and Y318 (RING1, helix H3) and G385 (tether following IBR domain and not visible in the crystal structure).

C Ubiquitination assays of Ubl domain mutants showing increased activity of interface mutants for both chain formation (top) and Miro1 ubiquitination (bottom), with a Coomassie-stained loading control shown in between. Time points as in Fig 1A. A non-specific, ATP-independent band is indicated (*).

I306 and E309 of the RING1 domain (Figs 3B and EV1). Comparison of this interface in UblR0RBR with that in R0RBR reveals a number of changes (Fig 3C). A salt bridge formed by R234 and E404 pinning the domains together is absent in the R0RBR structure, but present when the Ubl domain is attached. A hydrogen bond between H227 and E300 is observed in the presence of the Ubl domain, but not in the R0RBR structures (Fig 3C). In the Ubl-bound structure, the side chains of H302 and E300 (and H227) are facing into the interface, while in the R0RBR structures, these residues are flipped out ~180° away from the interface (Fig 3C). This latter conformation is

observed in all R0RBR structures, suggesting this is a consequence of the absence of the Ubl domain, rather than an artefact of crystal packing.

## A negative charge at position 65 of the Ubl domain causes remodelling of the RING1/RING0 interface

Phosphorylation of S65 in the Ubl domain enhances parkin activity (Kondapalli *et al*, 2012). In order to understand what conformational changes occur upon phosphorylation of the Ubl domain, we

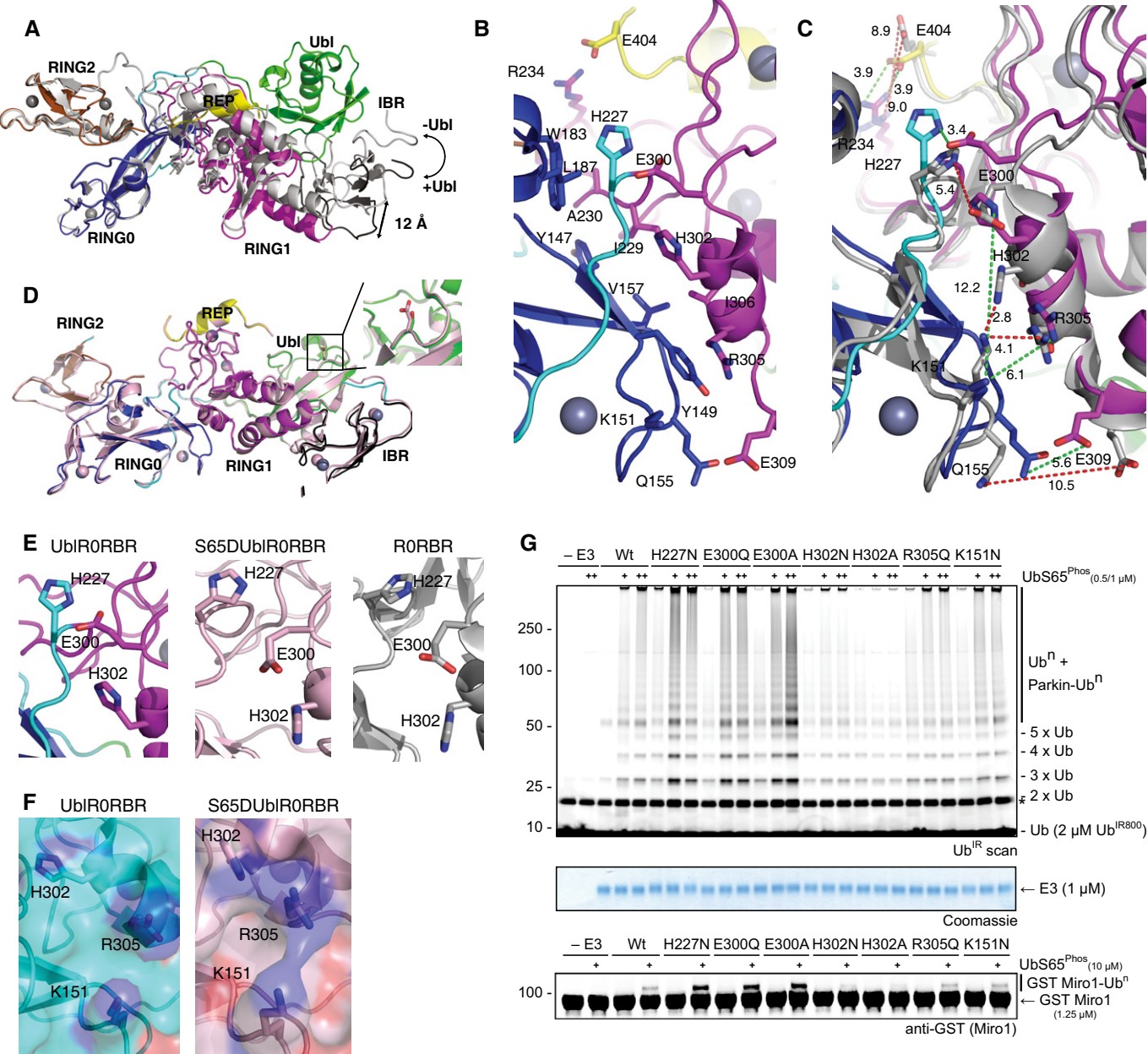

**Figure 3. The RING0/RING1 interface is a hinge that remodels upon parkin phosphorylation.**

A  Overlay of UblR0RBR (coloured according to Fig 1C) and R0RBR (grey) structures. Absence of the Ubl domain causes a hinge opening at the RING0/RING1 interface.

B  Close-up of the RING0/RING1 interface in UblR0RBR showing key residues that are involved in ionic/hydrogen bond interactions.

C  Overlay of the RING0/RING1 interface from UblR0RBR (coloured) and R0RBR (grey). Green dashes show distances in Å between residues in UblR0RBR, and red dashes show distances between residues in R0RBR.

D  Structure of S65DUblR0RBR (pink) overlaid with UblR0RBR parkin (coloured according to Fig 1C). The position of D65 or S65 is shown in inset.

E  Comparison of the hinge in UblR0RBR (left), S65DUblR0RBR (middle) and R0RBR (right).

F  Surface of UblR0RBR (left) and S65DUblR0RBR (right) showing continuous basic patch formed by hinge opening.

G  Ubiquitination assays of mutations in the RING0/RING1 interface. Mutations in residues that pin H302 in the flipped-in position are activating, mutation of the basic patch prevents parkin activity, even in the presence of increasing concentrations of phosphoubiquitin (+). This is the case for both chain formation (top) and Miro1 ubiquitination (bottom). A Coomassie-stained loading control is shown in between. A non-specific, ATP-independent band is indicated (*).

attempted unsuccessfully to crystallize phosphorylated parkin. Therefore, we crystallized a version of parkin carrying a negative charge at position 65 (S65DUblR0RBR). This protein, as well as

S65EUblR0RBR, showed increased ubiquitination activity compared to wild-type parkin that was further enhanced by phosphoubiquitin in a similar fashion to phosphorylated UblR0RBR (Appendix Fig S5).

The S65DUblR0RBR crystals grew under the same conditions, with the same unit cell dimensions and crystal packing, as the wild-type UblR0RBR crystals, suggesting little or no structural changes. The structure of S65DUblR0RBR parkin was refined to 2.4 Å (Table 1), with the same residues present in the density as in the wild-type UblR0RBR crystals. Superposition of S65DUblR0RBR and UblR0RBR (rmsd 0.58 Å) reveals no global conformational changes either in the Ubl domain, or in other parts of the protein (Fig 3D). However, close inspection reveals that H227, E300 and H302 of the RING0/RING1 interface are remodelled in the S65DUblR0RBR structure to resemble the interface observed in the truncated R0RBR structures (Fig 3E). Interestingly, the remodelling of these residues gives rise to a continuous basic patch comprising K151, H302 and R305 on the surface of Parkin, which is not present when the Ubl domain is unmodified (Fig 3F). Since parkin ubiquitination activity is further enhanced through phosphoubiquitin binding, we hypothesized that this basic patch is the phosphoubiquitin-binding site. To test this, we generated H302A parkin and assayed this for interaction with phosphoubiquitin (pUb) (Appendix Fig S6). In contrast to wild-type parkin, H302A parkin does not bind pUb. Furthermore, mutation of residues at the top of the interface that pin the E300 and H302 in the flipped-in conformation is more active than wild-type parkin, while mutation of the residues in the basic patch displays diminished activity compared to wild-type, and cannot be stimulated with pUb (Fig 3G). Taken together, these data suggest that the addition of a negative charge at position 65 of the Ubl domain alters the surface of parkin at the RING0/RING1 interface to facilitate optimal pUb binding.

## Phosphorylated ubiquitin binding induces a structural change in parkin

PINK1-mediated phosphorylation of ubiquitin at S65 (pUb) enhances parkin activity. In order to understand how this is achieved, we performed chemical shift perturbation experiments monitored by $^{1}$H-$^{15}$N TROSY spectroscopy. Addition of pUb to R0RBR results in a cluster of residues with significant chemical shift changes including Y149, C150, K151 and G152 (RING0); R275, F277, V278, D280, Q282, G284, Y285 and S286 (β16–β17 in RING1); and I306, G308, E309, Y312 and N313 (helices H2, H3 in RING1) (Fig 4A; Appendix Fig S7). These residues surround the hinge region at the RING0/RING1 interface that is optimized for pUb binding (Fig 3B, C, E and F). There are no significant chemical shift changes in the RING2 domain or RING0/RING2 interface (Fig 4A), indicating that pUb binding does not alter the interface between these two domains that includes the catalytic cysteine. These data reinforce that the hinge region at the RING0/RING1 interface is the pUb-binding site. Using these chemical shift perturbation experiments, we generated a model of parkin in complex with pUb (Fig 4B) using HADDOCK. The orientation of the pUb with respect to parkin was confirmed by using paramagnetic relaxation enhancement experiments with a spin label covalently attached to either L8C-pUb or K48C-pUb and observing residues with signal intensity changes in parkin in $^{1}$H-$^{15}$N TROSY spectra. In the model, pUb sits across a V-shaped cavity formed at the RING0/RING1 hinge region (Fig 4B). The basic triad, K151-H302-R305, surrounds the phosphoSer65 of ubiquitin. The pUb orientation is

governed by the β1–β2 loop in pUb that interacts with residues on the bent helix H3 of the RING1 domain and residues in the adjacent IBR domain. The C-terminal tail of pUb including residues V70 and L71 runs parallel to helix H3 of the RING1 domain. In addition, residues in the β3–β4 region of pUb (I44, A46) intercalate between strands β16–β17 and helix H3 of the RING1 domain. It is interesting that one component for pUb recognition, the short β16–β17 region in RING1, appears to be absent in all other RING domain protein structures. Further, the position of the β16–β17 loop is absent in the structure of the RBR E3 ligase HHARI (Duda *et al*, 2013) where it is replaced by a ubiquitin-associated like domain. The pUb-binding site is remote from the typical E2-binding site expected to reside on the opposite end of the RING1 domain.

Since the crystal structures of UblR0RBR and NMR titration experiments show that the RING0/RING1 interface is altered upon Ubl binding, we hypothesized that a similar allosteric event might occur through pUb binding to parkin. Consistent with this, several chemical shift changes occur in residues at the junction of the C-terminus of helix H3 and the IBR domain upon pUb addition (Fig 4A). The isolated IBR domain shows no chemical shift perturbations when S65EUb is titrated in, suggesting the chemical shift changes at the helix H3/IBR domain interface might result from a structural change that accompanies pUb binding. In order to test this, we used isothermal titration calorimetry (ITC) to measure thermodynamic changes in enthalpy (ΔH) and entropy (ΔS) that occur upon binding of R0RBR with pUb, S65DUb or S65EUb (Fig 4C, Table 2). As a control, we measured the binding of pUbl and phosphomimetic Ubl domains to R0RBR where remodelling of the RING0/RING1 interface is the most notable structural change. As expected from the crystal structures, pUbl interacts exothermically with R0RBR with negative enthalpy (−16 kJ/mol) and small entropy (33 J/mol˚K) changes consistent with new ionic and hydrophobic interactions formed and minimal overall structural change (Fig 4C and D). The data also show that Ubl binding to R0RBR is enthalpically driven regardless of whether the phosphomimetics (S65DUbl, S65EUbl) or pUbl is used, indicating the phosphomimetic Ubl and pUbl proteins have similar binding modes to R0RBR. Remarkably, the interaction of pUb or phosphomimetic ubiquitin (S65DUb, S65EUb) with R0RBR, or pUb with full-length parkin, has opposite enthalpy and entropy changes (Fig 4C and D, Table 2). In contrast to Ubl binding, the binding of pUb to R0RBR is an endothermic binding event marked by large positive enthalpy (+32 kJ/mol) and entropy (+261 J/mol˚K) changes (Fig 4C and D). The large positive entropy change signifies that the pUb–R0RBR interaction is driven by an increase in disorder in the system, a result of a loss of structure in parkin that accompanies pUb binding (Fig 4C). Interestingly, three-dimensional structures of the isolated IBR domain show little structure in the region that comprises the C-terminus of helix H3 of the RING1 domain, the site of several chemical shift changes upon R0RBR interaction with pUb (Beasley *et al*, 2007). This is in contrast to the crystal structures (Fig 1C) that show a well-defined, albeit bent helix that interacts with the first Zn$^{2+}$-binding site of the IBR domain and the C-terminus of helix H1 in the RING1 domain. Together, these data show that pUb binding to the RING0/RING1 hinge interface results in a concomitant loss of structure near the C-terminus of helix H3 (RING1) and the IBR domain.

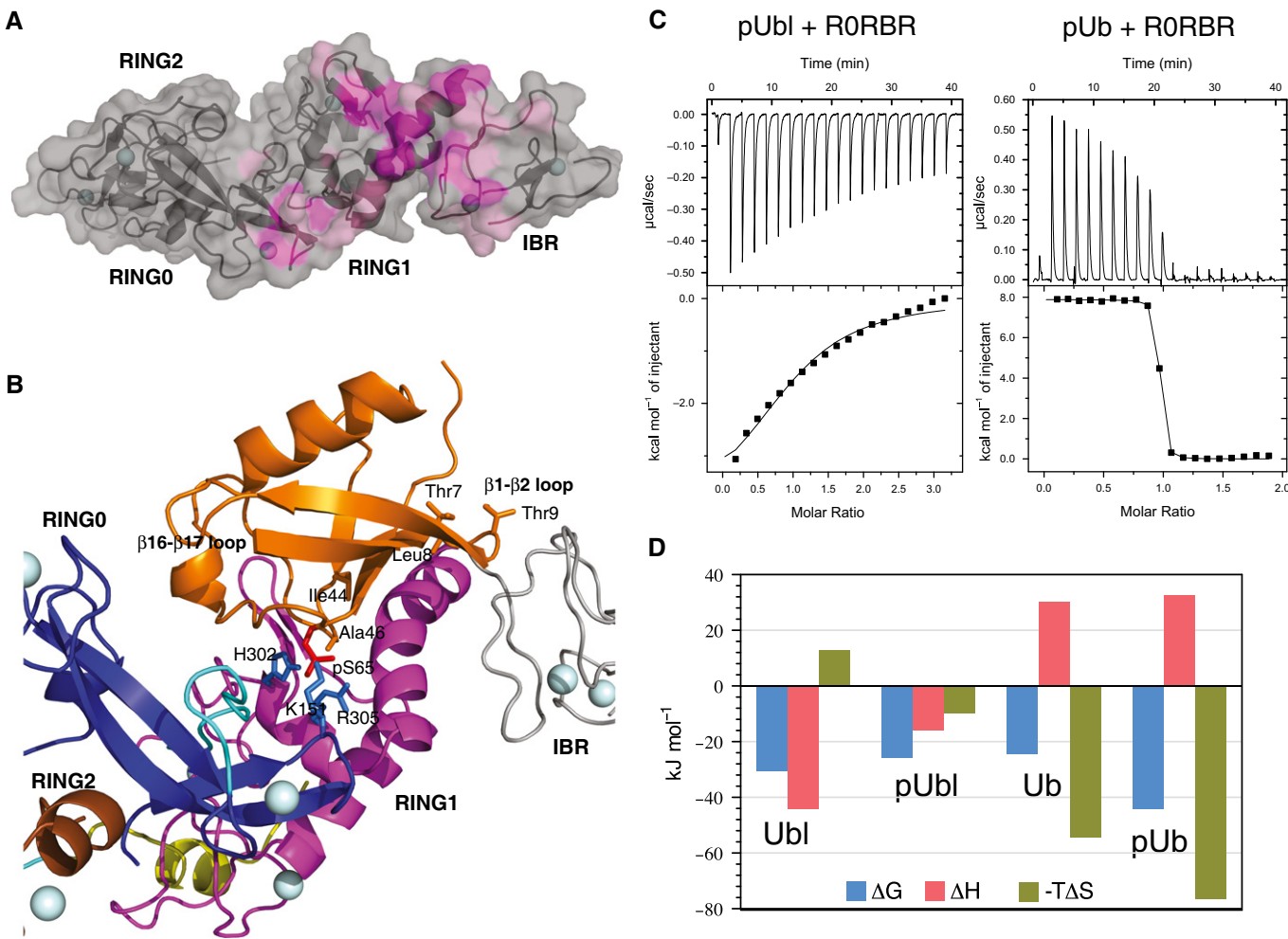

**Figure 4. Mechanism of phosphoubiquitin binding and conformational change in parkin.**

A   Chemical shift perturbation map of R0RBR upon the addition of pUb. A colour gradient is used to show residues and the corresponding surface with little or no changes (grey) and scaled from light pink (average chemical shift change) to magenta (greater than average + 2 standard deviations). Parkin is shown as a transparent surface over a ribbon, and the orientation of the molecule is similar to that shown in Fig 1C (bottom).

B   Representative model of pUb interaction with parkin derived from HADDOCK calculations using chemical shift perturbation and paramagnetic relaxation enhancement experiments. One hundred water-refined structures were calculated that all showed a similar pUb orientation. The best twenty complexes showed excellent agreement (backbone rmsd 0.37 Å, pUb + R0RBR). The model shown has been rotated 90° (x) and 20° (y) with respect to Fig 1C (top). The colours used are the same as in Fig 1C with pUb shown in orange. The pS65 is shown as red sticks, and the basic triad consisting of K151, H302 and R305 are shown as blue sticks. Several important residues that had significant chemical shift changes in pUb upon R0RBR interaction are indicated.

C   Representative isothermal titration calorimetry experiments showing the exothermic binding of pUbl to R0RBR parkin (left) and endothermic binding of pUb to R0RBR parkin (right).

D   Bar graph showing the thermodynamic properties for Ubl, pUbl, Ub and pUb binding to R0RBR. The data show Ubl binding and pUbl binding are driven by enthalpy (ΔH) changes, while Ub binding and pUb binding are driven by entropy changes (ΔS) indicative of a pUb-induced conformational change.

## Phosphorylated ubiquitin binds to parkin and displaces the Ubl domain

Our crystallographic studies show that phosphorylation of the Ubl domain optimizes the RING0/RING1 interface for pUb binding, while NMR and ITC experiments show pUb binding causes a loss of structure near the IBR domain. These allosteric structural changes infer that pUbl and pUb cannot be bound to parkin simultaneously. In order to test this, we performed a competition experiment monitored by NMR spectroscopy. We titrated R0RBR parkin into [13]C-labelled S65EUbl (Fig 5A, top) and followed this in

[1]H-[13]C-HMQC spectra such that signals from both the unbound and bound forms of S65EUbl were visible. For example, changes in the positions of A46 and L61 of the Ubl domain show a clear interaction with R0RBR. Upon addition of S65EUb, the signals for the bound form of S65EUbl revert back to those of the unbound S65EUbl state (Fig 5A, bottom). This experiment shows that phosphomimetic Ub binding displaces the bound S65EUbl domain from parkin. In a reciprocal experiment, we titrated R0RBR parkin into [13]C-labelled S65EUb to form the activated R0RBR–S65EUb complex (Fig 5B, top). Upon addition of S65EUbl, the signals for the bound form of S65EUb remain (Fig 5B, bottom) despite S65EUb and S65EUbl

**Table 2.  Thermodynamic properties for parkin activation.**

| Protein | Activator | $K_d$ (µM) | N | ΔH (kJ/mol) | ΔS (J/molK) |
|---|---|---|---|---|---|
| A. R0RBR Parkin with Ubl and Ub | | | | | |
| R0RBR | Ubl | 3.8 ± 1.3 | 0.98 | −44 ± 3 | −44 |
| | Ubl S65D | 5.0 ± 0.9 | 1.03 | −38 ± 1 | −29 |
| | Ubl S65E | 5.3 ± 1.2 | 0.98 | −46 ± 2 | −54 |
| | pUbl | 26 ± 10 | 1.04 | −16 ± 9 | 33 |
| R0RBR | Ub | 67 ± 1.0 | 1.05 | 31 ± 1 | 184 |
| | Ub S65D | 6.6 ± 0.3 | 1.01 | 25 ± 1 | 184 |
| | Ub S65E | 6.6 ± 0.1 | 1.10 | 48 | 261 |
| | pUb | 0.016 ± 0.002 | 0.93 | 32 | 261 |
| B. Full-length Parkin with Ubl and Ub | | | | | |
| Parkin | Ubl | N.O. | – | – | – |
| | Ub | 45 ± 9 | 0.90 | 10 ± 1 | 115 |
| | pUb | 0.16 ± 0.02 | 1.09 | 50 | 293 |
| | | 0.37 ± 0.04[a] | – | – | – |
| pParkin | pUb | 0.021 ± 0.006 | | | |
| | | 0.017 ± 0.005[a] | – | – | – |

Average values from a minimum of two experiments.
[a]Data from Ordureau *et al* (2014).
N.O.,binding not observed.

having similar dissociation constants for R0RBR (Table 2). This indicates that phosphomimetic Ubl domain is unable to bind to an activated R0RBR–S65EUb complex. Together, these experiments show that simultaneous binding of S65EUbl and S65EUb to R0RBR is not possible. Further, the data show that although S65EUbl optimizes the pUb-binding site, the binding of S65EUb fully releases S65EUbl from parkin, consistent with the proposed allosteric loss of structure to the C-terminus of helix H3 and IBR domain that would interfere with the Ubl-binding site. This would also indicate that activation of parkin through pUb binding would prevent re-engagement with the Ubl domain until the pUb is released.

**Activation by phosphoubiquitin uncovers a ubiquitin recognition site**

The residues involved in the Ubl/RING1 interface are largely conserved between ubiquitin and the Ubl domain (Fig 6A and B). Thus, displacement of the Ubl domain by phosphoubiquitin binding may lead to exposure of a surface competent for interaction with ubiquitin. We and others have previously reported an interaction between active parkin and ubiquitin (Chaugule *et al*, 2011; Zheng & Hunter, 2013). Therefore, we wondered if the consequence of Ubl displacement is to provide a ubiquitin-binding surface for recruitment of the charged E2~ubiquitin complex. In order to test this hypothesis, we generated mutations in helix H1 of the RING1 domain and assayed them for ubiquitin chain formation and Miro1 ubiquitination. Single-point mutations D262A, T270R and D274R are deficient in both ubiquitin chain formation and Miro1 ubiquitination (Fig 6C), in contrast to corresponding Ubl domain mutations that are activating (Fig 2C). Interestingly, the single L266K mutant is active for ubiquitin chain formation but deficient in Miro1 ubiquitination. This suggests that although L266K would be expected to

repel the Ubl domain, it is still able to form a productive complex with the E2~Ub. In contrast, a triple mutant along the Ubl-binding surface of helix 1, D262A/L266K/D274R, is unable to form ubiquitin chains or ubiquitinate Miro1 (Fig 6C). Recent data suggest that in the presence (but not absence) of phosphoubiquitin, parkin forms a discrete complex with E2~Ub (Ordureau *et al*, 2015). Therefore, we assayed the ability of the wild-type phosphoparkin/phosphoubiquitin complex to interact with E2, or E2~Ub (Fig 6D). Phosphoparkin and phosphoubiquitin form a complex as measured by size-exclusion chromatography consistent with the tight binding observed in ITC experiments (Table 2). Addition of the E2 UbcH7 does not produce a further shift consistent with the observed moderate binding affinity of UbcH7 for phosphoparkin in the presence of phosphoubiquitin ($K_d$ = 19 µM, Fig 6E). In contrast, the addition of UbcH7~Ub leads to the formation of a higher molecular weight complex that has an apparent 20-fold tighter affinity than UbcH7 alone (Fig 6D and E), suggesting the ubiquitin molecule is recognizing an additional binding site on parkin. To test whether this ubiquitin site includes the D262/L266/D274 residues along helix 1, similar size-exclusion chromatography experiments were conducted using the D262A/L266K/D274R triple mutant protein. In contrast to wild-type parkin, the RING1 triple mutant is unable to form the pParkin/pUb/UbcH7~Ub complex (Fig 6D). Taken together, these data suggest that the consequence of Ubl displacement is the creation of a ubiquitin-binding site on RING1 to facilitate E2~Ub recruitment.

# Discussion

Until 2011, parkin was thought to be constitutively active. Since then, it has become clear that parkin is regulated in multiple

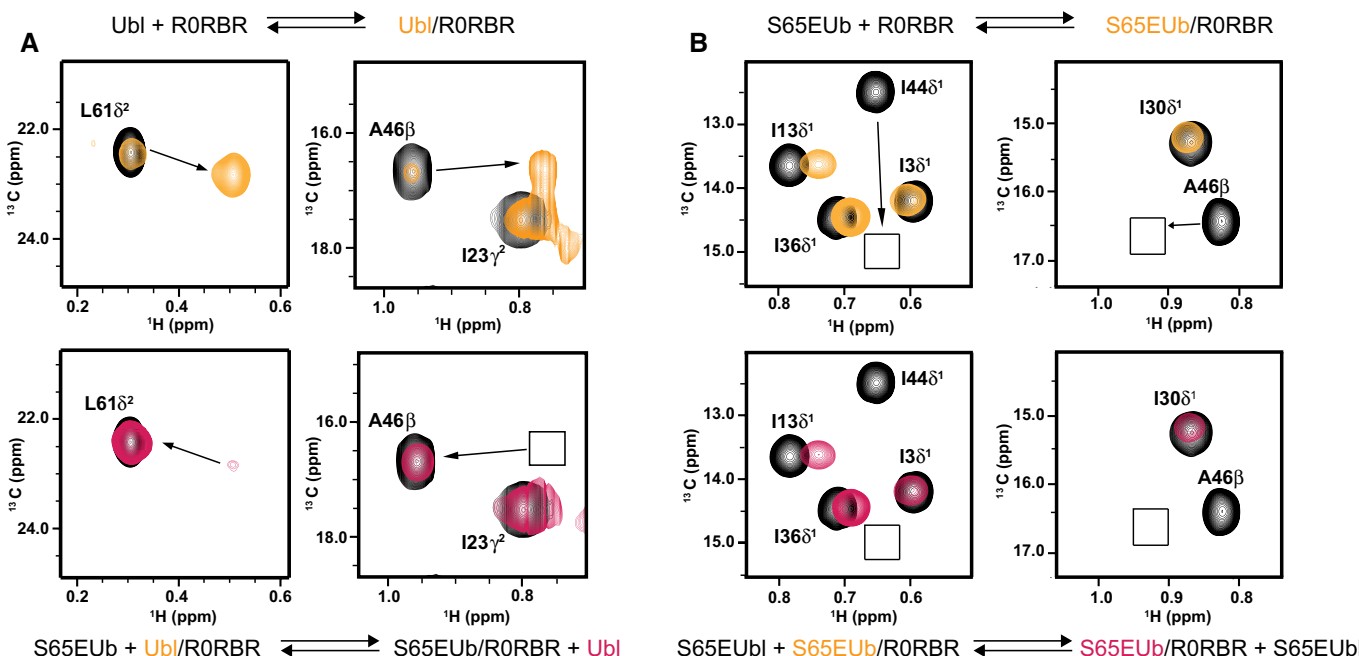

**Figure 5.  pUb binding displaces the Ubl domain.**

A  (Top) Selected regions of $^1$H-$^{13}$C HMQC spectra for 150 μM $^{13}$C-labelled S65EUbl (black contours) following the addition of one equivalent of unlabelled R0RBR parkin. Approximately 83% of the S65EUbl is bound to R0RBR parkin (orange contours) based on a $K_d$ = 5 μM (Table 2). The arrows show the position of the bound $^{13}$C-labelled S65EUbl signal upon R0RBR binding. (Bottom) The same sample and regions following the addition of 150 μM unlabelled S65EUb show the disappearance of most of the bound $^{13}$C-labelled S65Ubl-R0RBR species and re-appearance of unbound S65EUb (red contours), indicating S65EUb is able to displace S65EUbl.

B  (Top) Selected regions of $^1$H-$^{13}$C HMQC spectra for $^{13}$C-labelled S65EUb (black contours) following the addition of one equivalent of unlabelled R0RBR parkin (orange contours). Boxes show the position of the bound $^{13}$C-labelled S65EUb signal upon R0RBR binding visible at lower contour levels. (Bottom) The same sample following the addition of one equivalent unlabelled S65EUbl shows no change in the intensities and position of the bound $^{13}$C-labelled S65EUb (red contours), indicating that S65EUbl is unable to bind to R0RBR in the presence of S65EUb.

ways. The Ubl domain inhibits parkin activity (Chaugule *et al*, 2011), the REP element blocks an E2-binding site (Trempe *et al*, 2013), and it has been proposed that the catalytic cysteine, C431, is occluded by the RING0 domain (Riley *et al*, 2013; Trempe *et al*, 2013; Wauer & Komander, 2013). An important observation from the structures of UblR0RBR and S65DUblR0RBR parkin is that the catalytic cysteine (C431) in the RING2 domain remains in a similar environment as that found in truncated parkin structures lacking the Ubl domain. Further, NMR chemical shift perturbation experiments show minimal changes for C431 or to residues at the RING0/RING2 interface near the catalytic site, indicating this region of parkin does not undergo any large structural change that might expose C431. Interestingly, analysis of the UblR0RBR, S65DUblR0RBR and R0RBR structures all show the thiol group of C431 is partially exposed and capable of accepting a ubiquitin molecule from the E2~Ub conjugate. Thus, the increased ubiquitination reactivity observed upon phosphoubiquitin binding likely results from enhanced recruitment of the E2~Ub conjugate due to exposure of an E2~Ub-binding site following release of the Ubl domain (Fig 6), rather than a change in structure near the catalytic C431 site.

The discovery of autoinhibition propelled efforts to uncover activators of parkin, the most compelling of which is the kinase, PINK1, upstream of parkin (Clark *et al*, 2006; Park *et al*, 2006). PINK1 phosphorylates the Ubl domain and ubiquitin at the equivalent residue S65, leading to parkin activation (Kondapalli *et al*,

2012; Kane *et al*, 2014; Kazlauskaite *et al*, 2014b; Koyano *et al*, 2014). Recent studies report effects of each phosphorylation signal on the activity of parkin. For example, while parkin can be phosphorylated by PINK1 in the absence of phosphoubiquitin, the addition of pUb enhances parkin phosphorylation, suggesting that parkin bound to pUb is a better substrate for PINK1 than parkin alone (Kazlauskaite *et al*, 2015). However, phosphorylation of parkin is independent of parkin catalytic activity and parkin associates tightly with ubiquitin chains only when both ubiquitin and parkin are phosphorylated (Ordureau *et al*, 2014), leading to the proposal of a feed-forward mechanism. Furthermore, in cells where the only ubiquitin source is a non-phosphorylatable S65A mutant, parkin is still phosphorylated (Ordureau *et al*, 2015), albeit GFP-tagged parkin, which is an important caveat since it is not established that N-terminally tagged parkin species can be interpreted as wild-type (Chaugule *et al*, 2011; Burchell *et al*, 2012). We present here a mechanistic model of parkin inhibition and allosteric activation based on our structural, biochemical and biophysical insights (Fig 7). First, the Ubl domain inhibits parkin activity in the absence of phosphoubiquitin signal (Fig 7, step 1). In this inhibited conformation, the E300 and H302 are facing into the RING0/RING1 interface. Upon activation, PINK1 phosphorylates ubiquitin and the Ubl domain of parkin (Fig 7, step 2) to induce a remodelling of the RING0/RING1 interface (Fig 3C–E). Although phosphorylation of ubiquitin alone leads to a dramatic increase in its affinity for parkin (Table 2, Fig 4D), this optimized

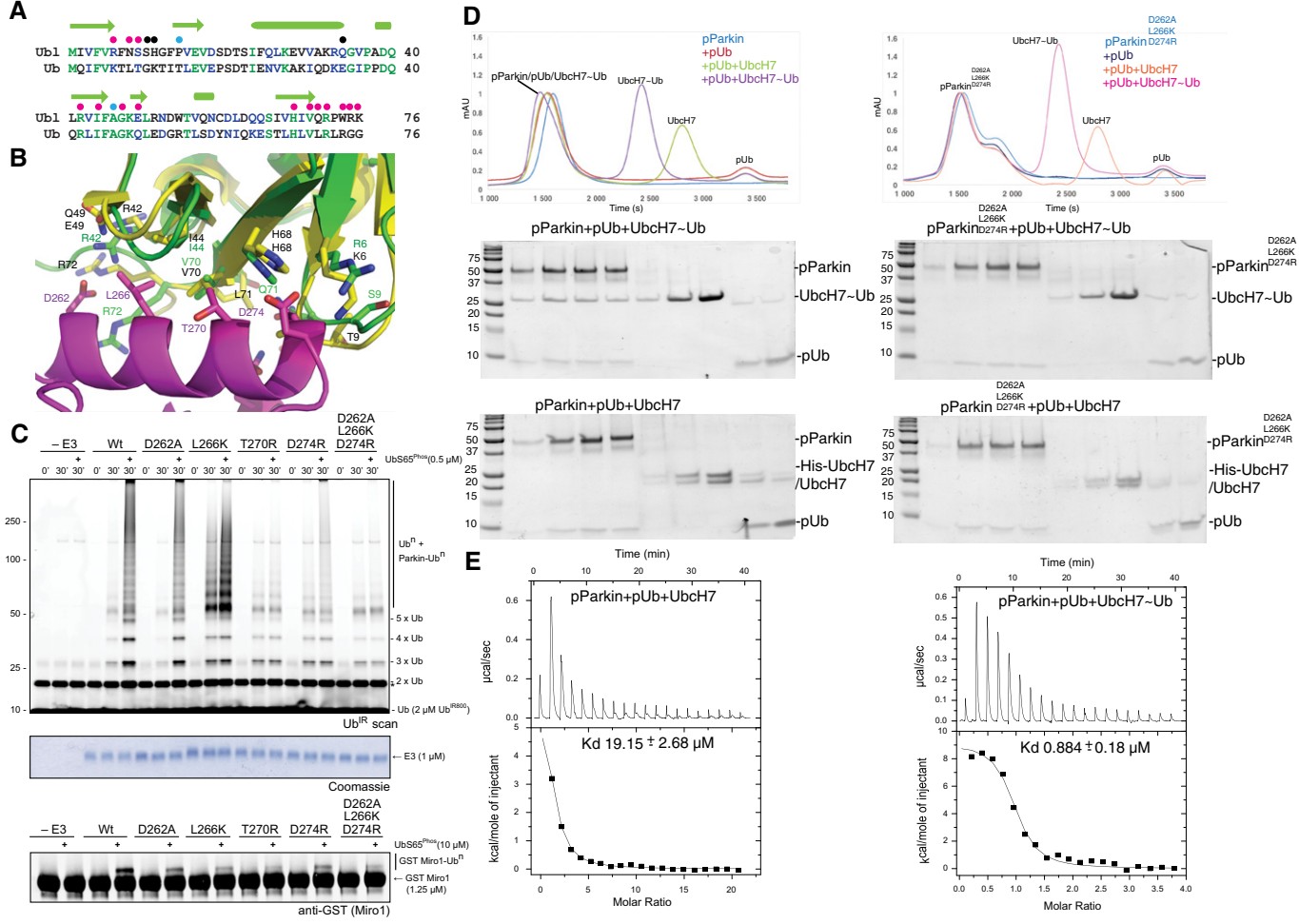

**Figure 6. pUb binding to parkin uncovers a ubiquitin binding site.**

A   Structure-based alignment of Ubl domain and ubiquitin. Conserved residues are shown in green, and conservative substitutions are shown in blue. Residues that interact with the RING1 domain (magenta), IBR (black) and tether (cyan) are denoted with circles.

B   Close-up of the interface between helix H1 in the RING1 domain and the Ubl domain based on the structure shown in Fig 1C. Key residues for the interaction are highlighted. The structure of ubiquitin is superimposed with Ubl domain including the corresponding residues found at the RING1 interface.

C   Ubiquitination assay of RING1 mutations in the Ubl/RING1 interface. Effects of single-point mutations and a triple mutation on both ubiquitin chain formation (top) and Miro1 ubiquitination (bottom) are shown. A Coomassie-stained loading control is shown in between. A non-specific, ATP-independent band is indicated (*).

D   Size-exclusion chromatography of phosphoparkin (pParkin), phosphoparkin and phosphoubiquitin (pUb), phosphoparkin, phosphoubiquitin and UbcH7, and phosphoparkin, phosphoubiquitin and UbcH7 isopeptide linked to ubiquitin (UbcH7~Ub) (left) versus the same experiment using the triple mutant of parkin (right). A coloured key of each trace is provided, and Coomassie-stained gels of the indicated peaks are shown below the trace.

E   Isothermal titration calorimetry of phosphoparkin complexed with phosphoubiquitin titrated with either UbcH7 alone (left) or UbcH7~Ub (right). Dissociation constants for each titration are shown.

parkin conformation exhibits a further 10-fold greater affinity for pUb (Ordureau *et al*, 2014) (Table 2) mediated by the H302/R305/K151 basic patch (Fig 7, step 3, Fig 3B). Our model of the pUb-binding site along helix H3 of RING1 is confirmed by the recent crystal structure of pUb bound to *Pediculus humanus* R0RBR parkin (Wauer *et al*, 2015). Phosphoubiquitin binding leads to an entropically driven loss of structure in the RING1 bent helix (H3) (Fig 7, step 3). This remodelling leads to release of the Ubl domain, exposing the binding surface on RING1. Since the residues in the Ubl domain that bind the RING1 domain are well conserved with ubiquitin (Fig 6A and B), our model predicts that this exposed RING1 surface will engage ubiquitin in an E2~Ub conjugate to form a 'charged' parkin (Fig 7, step 4). This accounts

for observations that parkin can function with different E2s. Finally, the tightly associated inhibited structure, small conformational changes, interface remodelling and multiple binding sites explain why different patient mutations have different effects on parkin activity. Although there is some debate on the 'order' of activating events, our study supports a model whereby each phosphorylation event enhances the activation of parkin, which is likely a major mechanism of parkin regulation in the context of low concentrations of pUb in cells (Koyano *et al*, 2014). The mechanistic model presented here provides important insights into parkin regulation that will be key to designing therapeutics that can control the activity of parkin and may be useful in the treatment of early-onset forms of Parkinson's disease.

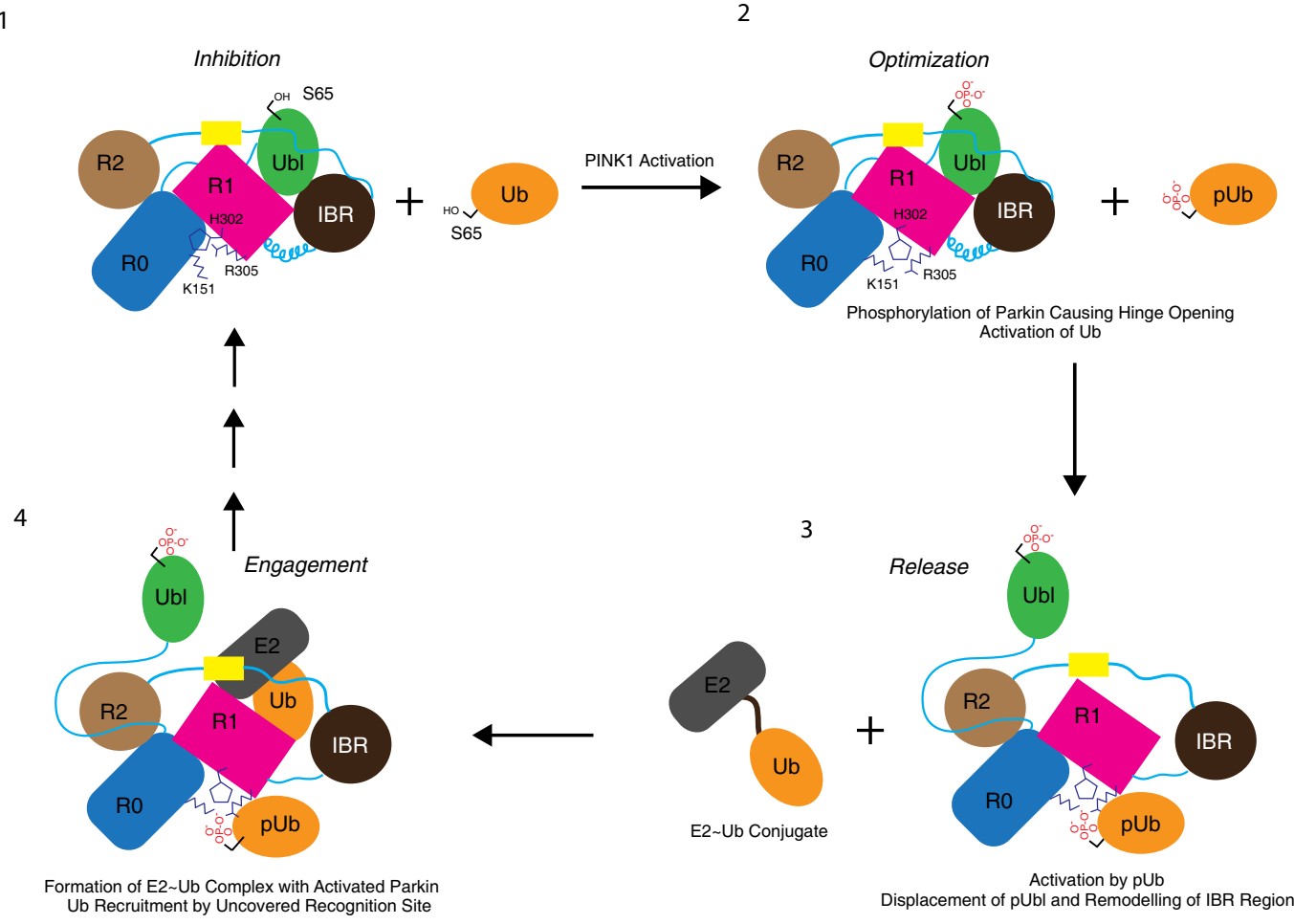

**Figure 7.  Model of parkin inhibition and activation.**
(1) Inhibition: Wild-type parkin is autoinhibited in the absence of phosphoubiquitin. PINK1 activation leads to phosphorylation of the accessible S65 in both parkin and ubiquitin. (2) Optimization: Parkin phosphorylation stabilizes the flipped-out conformation of H302, thus optimizing the phosphate-binding site. (3) Release: Phosphoubiquitin binds to helix H3 of the RING1 domain, leading to displacement of the Ubl domain and loss of structure near the RING1/IBR interface. (4) Engagement: The ubiquitin and E2 binding surfaces uncovered by displacement of the Ubl domain engage with charged E2~Ub conjugate poised for ubiquitin transfer.

# Materials and Methods

### Protein purification and crystallization

UblR0RBR Parkin (residues 1–83, 144–465), phosphomimetic S65DUblR0RBR and R0RBR (residues 141–465, lacking Ubl domain and linker) constructs were expressed as His-Smt3 fusion proteins in BL21 *E. coli* cells as previously described (Chaugule *et al*, 2011; Spratt *et al*, 2013). Media were supplemented with 50 μM ZnCl$_2$, and cells were induced at 0.6 OD$_{600}$ with 120 μM of isopropyl β-D-1-thiogalactopyranoside (IPTG) and incubated overnight at 16°C. Cells were lysed, and protein was purified over Ni-NTA resin. Beads were incubated overnight at 4°C with ULP1 protease to remove the His-Smt3 tag. The flow through from Ni-NTA incubation was collected and passed through a Mono-Q column and then loaded on Superdex-75 equilibrated with Buffer A (20 mM Tris pH 7.5, 75 mM NaCl and 250 μM TCEP).

Fractions containing purified protein were pooled and concentrated using Viva spin 10-kDa cut-off concentrators. UblR0RBR (6 mg/ml) and S65DUblR0RBR (2.8 mg/ml) parkin were used for sitting drop crystallization using commercial screens. Protein crystals were obtained for both parkin forms at 4°C using 200 mM LiSO$_4$, 100 mM BIS-TRIS pH 5.5 and 20% PEG 3350 by sitting drop vapour diffusion method. Crystals appeared in 2–3 days and then allowed to grow for a week. Crystals were flash-frozen using 20% glycerol as cryoprotectant. Native data sets were collected at Diamond Light Source, beamline IO4-1 and IO4.

### Synthesis and purification of phosphorylated ubiquitin and Ubl domain

*Pediculus humanus* GST-PINK1 (128-C) was purified on a GSTrap FF column (GE Healthcare) using 50 mM Tris, 250 mM NaCl, 1 mM DTT at pH 7.5 and eluted with freshly prepared 50 mM Tris,

250 mM NaCl, 1 mM DTT and 10 mM glutathione at pH 7.5. GST-PINK1 (10 μM) was incubated with either Ub (500 μM) or UblD (500 μM) and dialysed against buffer containing 50 mM Tris, 0.5 mM DTT, 10 mM $MgCl_2$ and 5 mM ATP (pH 7.5) for 2 h at 25°C. The reaction was monitored to completion using Phos-Tag™ SDS–PAGE. Following the reaction, GST-PINK1 was removed from the reaction using a GSTrap FF column, collecting the flow-through fractions containing pUb or the pUbl domain. The phosphorylated proteins were separated from unphosphorylated species on a HiTrap Q column using 20 mM Bis-Tris propane (pH 8.7) and a 2-h 0–100% 0.5 M NaCl elution gradient. Unphosphorylated species were found in the flow through, while pure pUb and pUblD eluted around 20–40% of the salt gradient.

### Synthesis and purification of UbcH7~Ub

The isopeptide-linked UbcH7~Ub conjugate was synthesized following the method of Plechanovova *et al* (Plechanovova *et al*, 2012). His-tagged Uba1 (25 μM, not TEV-cleavable), His-TEV-Ub (200 μM) and UbcH7 (C17S/C86K/C137S, 400 μM) were incubated at 37°C for up to 16 h in 50 mM CHES, 150 mM NaCl, 10 mM ATP and 10 mM $Mg^{2+}$ at pH 9.0 until UbcH7~Ub conjugate formation reached a stable level as viewed by SDS–PAGE electrophoresis. Purification of the UbcH7~Ub conjugate was performed with a HisTrap column (GE Healthcare) initially to remove UbcH7, and following TEV cleavage to remove His-Uba1. Purified UbcH7~Ub was isolated using a HiLoad Superdex (GE Healthcare) size-exclusion column to separate UbcH7-Ub from Ub.

### Structure determination

Data were indexed, processed and scaled using Mosflm and Scala as implemented in CCP4 (CCP4, 1994). The UblR0RBR structure was determined by molecular replacement. Full-length rat parkin structure (4K95) was used as a search model. The initial model was further built and refined using coot (Emsley & Cowtan, 2004) and autobuster (Bricogne *et al*, 2011). The phosphomimetic S65DUblR0RBR structure was solved using UblR0RBR as the search model, and refined as UblR0RBR. Both models have excellent geometry with statistics listed in Table 1.

### Isothermal titration calorimetry

All calorimetry experiments were performed using either a Microcal VP-ITC system (GE Healthcare) or a NanoITC (TA Instruments) at 25°C. All experiments were completed 2–3 times using freshly prepared proteins in either 50 mM HEPES, 50 mM NaCl and 250 μM triscarboxyethylphosphine (TCEP) at pH 7.5 or 50 mM HEPES, 150 mM NaCl and 0,5 mM TCEP at pH 8.0. The concentrations of each protein in the experiments were as follows: 50 μM R0RBR parkin titrated with 1.25 mM ubiquitin, 750 μM S65D ubiquitin, 400 μM phosphoubiquitin, 750 μM Ubl domain and 1 mM phosphoUbl domain. Full-length parkin (50 μM) was titrated with 1.75 mM ubiquitin, 1.75 mM S65D ubiquitin or 400 μM phosphoubiquitin. Phosphoparkin (50 μM) was titrated with 400 μM phosphoubiquitin. Phosphoparkin (35 μM) was titrated with 2.4 μM UbcH7 or UbcH7~Ub. Data were analysed using single-site binding models.

### Ubiquitination assays

Ubiquitination reactions were performed at 30°C in 50 mM DL-malic acid, MES monohydrate, Tris (MMT) pH 7.5, 50 mM NaCl, 2.5 mM $MgCl_2$, 5% (v/v) glycerol and 0.75 mM dithiothreitol (DTT) buffer system. All reactions contained 25 nM of recombinant human E1, 500 nM of UbE2L3 (UbcH7), 1 μM E3 and 5 mM ATP in a 20 μl final reaction volume.

Complete ubiquitination profiles were analysed using fluorescently labelled ubiquitin ($Ub^{IR800}$). Ubiquitin (residues 2–76) was expressed and purified bearing a GPLCGS overhang at the N-terminus. The cysteine residue in the overhang was targeted for site-specific incorporation of a DyLight™ 800 Maleimide (Life Technologies, 46621) dye following the manufacturer's protocol. Labelled species were purified by cation exchange chromatography and stored at −20°C as single-use aliquots. Reactions to monitor ubiquitination profiles contained 2 μM of $Ub^{IR800}$ and supplemented with either phosphorylated ubiquitin or unlabelled ubiquitin as indicated. Reactions were terminated with SDS loading buffer, resolved by SDS–PAGE and analysed by direct fluorescence monitoring using Li-COR® Odyssey Infrared Imaging System.

Glutathione S-transferase (GST)-human Miro1 (residues 1 to 592) was expressed and purified using standard protocols. GST-Miro1 ubiquitination reactions were carried out as above except with a final ubiquitin concentration of 20 μM (inclusive of indicated amount phosphorylated ubiquitin). Reactions were subjected to immunoblotting using anti-GST rabbit monoclonal primary antibody (1H14L28, Life Technologies, 1/3,000 dilution) and fluorescent-labelled secondary antibody (926-32213, Li-COR®, 1/10,000 dilution). Blots were visualized using Li-COR® Odyssey Infrared Imaging System.

### Size-exclusion chromatography–multiAngle light scattering analysis

Size-exclusion chromatography and multiangle light scattering (SEC–MALS) experiments were performed on a Dionex Ultimate 3000 HPLC system with an inline Wyatt miniDAWN TREOS MALS detector and Optilab T-rEX refractive index detector. SEC–MALS experiments were performed on a Superdex S75 CL 10/300 (GE Healthcare) column in buffer containing 50 mM HEPES with 100 mM NaCl at pH 7.5. Phosphoubiquitin (140 μM) and phospho-parkin proteins (20 μM) were incubated for 30 min before injections. Molar masses spanning elution peaks were calculated using ASTRA v6.0.0.108 (Wyatt). For the size-exclusion analysis with UbcH7 or UbcH7~Ub, phosphoparkin proteins (20 μM) were incubated with phosphoubiquitin (40 μM) and either UbcH7 (40 μM) or UbcH7~Ub (40 μM) for 30 min prior to injections.

### NMR spectroscopy

Uniformly $^2H,^{13}C,^{15}N$-labelled or $^2H,^{12}C,^{15}N$-labelled parkin R0RBR was expressed in minimal media supplemented with $^{15}NH_4Cl$ (1 g) and either $^2H,^{13}C$-glucose (2 g) or $^2H,^{12}C$-glucose (2 g)(Cambridge Isotope Laboratories). Cells were initially grown in 70% $D_2O$ until the $OD_{600}$ reached 0.9 units. Cells were pelleted, re-suspended in minimal media containing 99.8% $D_2O$ supplemented with 0.25 mM $ZnCl_2$ and grown until the $OD_{600}$ reached 0.8 units. Following the

addition of a further aliquot of ZnCl$_2$ (0.25 mM), the temperature was lowered to 16°C and cells were induced with 100 μM IPTG for 20 h. Assignment of R0RBR was accomplished using $^2$H,$^{13}$C,$^{15}$N-labelled protein (300–500 μM) in 25 mM HEPES, 100 mM NaCl, 0.5 mM TCEP and 90% H$_2$O/10% D$_2$O at pH 7.0 using DSS as an internal reference. All NMR data were collected at 25°C on a Varian Inova 600 MHz NMR spectrometer equipped with a triple resonance cryogenic probe and z-field gradients or an 850 MHz Bruker Avance III spectrometer with TCI cryoprobe. All $^1$H,$^{15}$N-HSQC spectra were collected in TROSY mode (Pervushin *et al*, 1997). Triple resonance pulse sequences (HNCA, HN(CO)CA, HN(CA)CB, HN(COCA)CB, HNCO, HN(CA)CO) (Yang & Kay, 1999) were collected in TROSY mode using deuterium decoupling as described for fully deuterated proteins. Heteronuclear NOE NMR experiments (Farrow *et al*, 1994) were performed at 600 MHz using 500 μM uniformly deuterated parkin 141-C. Proton saturation was achieved through a 5-s irradiation following an 11-s relaxation delay. The equivalent non-saturated experiment contained a 16-s relaxation delay. Both saturated and non-saturated experiments were conducted in duplicate. All data were processed using NMRPipe and NMRDraw (Delaglio *et al*, 1995) and analysed using NMRViewJ (Johnson & Blevins, 1994).

Protein interaction experiments were conducted using 100–200 μM $^2$H,$^{12}$C,$^{15}$N-labelled R0RBR and either $^{13}$C-labelled or $^2$H,$^{13}$C-labelled Ub or UblD proteins. $^1$H,$^{15}$N-TROSY spectra were collected to monitor chemical shifts of backbone amides in R0RBR. $^1$H,$^{13}$C-HMQC spectra (Tugarinov *et al*, 2004) were collected to monitor chemical shifts of side chain methyl groups in Ub or UblD. Chemical shift perturbations were quantified using the following weighted formula: $((0.2 \times \Delta\delta N^2) + \Delta\delta H^2)^{1/2}$ and plotted as a function of residue using Prism5 (Graphpad).

**Molecular modelling of phosphoubiquitin bound to parkin R0RBR**

Docking of pUb with R0RBR used chemical shift perturbations from NMR experiments (described above). Interacting residues in R0RBR were defined as those that shifted greater than the average shift + 0.5 standard deviations and had greater than 20% side chain accessible surface area. Due to the limited number of methyl-containing residues, residues in pUb were chosen where methyl resonances shifted regardless of accessible surface area. An upper distance limit of 5.0 Å was set for ambiguous distance restraints. A single unambiguous distance restraint of 3.0 Å was set based on mutagenesis experiments that indicated pS65 (Ub) was near H302 (R0RBR) that was also supported by chemical shift changes of neighbouring residues. The coordinates for S65DUblR0RBR was used as the starting structure following removal of the coordinates corresponding to the Ubl domain and adjoining linker region. Missing regions, where chemical shift changes were observed upon pUb interaction, were modelled in using the Modeller (Eswar *et al*, 2006) plug-in for UCSF Chimera (Pettersen *et al*, 2004). Docking of pUb with R0RBR used HADDOCK (Dominguez *et al*, 2003). A total of 1,000 initial complexes were calculated, and the best 100 structures were water refined. Standard parameters were used except inter_rigid (0.1) which was set to allow tight packing of the two proteins, and the unambiguous force constants were increased by five-fold compared to the unambiguous constants. The location and orientation of the pUb with respect to R0RBR were similar in all 100 water-refined

complexes, having a backbone rmsd = 0.37 ± 0.02 for the best 20 complexes.

Paramagnetic relaxation enhancement experiments (PRE) were used to confirm the orientation of the pUb molecule with respect to the R0RBR protein. Briefly, $^{12}$C/$^{14}$N/$^2$H-labelled cysteine-substituted Ub (L8C or K48C) was expressed in minimal media, purified and phosphorylated using *Pediculus humanus* GST-PINK1 as described above. Proteins were spin-labelled with 3-(2-iodoacetamido)-PROXYL (Sigma Aldrich) by incubating each protein with 10 molar equivalents for 2 h at room temperature and dialysed exhaustively at 4°C. ESI-MS confirmed quantitative (> 95%) phosphorylation and successful introduction of spin label (~90 and ~60% spin label incorporation for L8C and K48C, respectively). $^1$H-$^{15}$N TROSY spectra were collected monitoring 400 μM $^{13}$C/$^{15}$N/$^2$H-R0RBR before and after the addition of 1.2 molar equivalents of each nitroxide-labelled pUb variant. Identical spectra were collected following the addition of 10 molar equivalents of ascorbic acid to quench the nitroxide spin tag. PRE factors in R0RBR were determined in NMRViewJ by measuring peak intensities with an active and quenched nitroxide spin tag.

**Data and reagents availability**

The PDB files of the structures determined in this study have been deposited with the RCSB Protein Data Bank (www.rcsb.org) with accession codes 5C1Z (UblR0RBR) and 5C23 (S65DUblR0RBR). NMR assignments have been deposited to the BMRB (www.bmrb.wisc.edu) under accession codes 25708 (S65EUb), 25707 (Ub bound to R0RBR parkin), 25709 (S65EUb bound to R0RBR parkin) and 26605 (Ubl domain bound to R0RBR parkin). Recombinant proteins and plasmids generated for the present study are available to request on the MRC-PPU reagents website (https://mrcppureagents.dundee.ac.uk/).

**Expanded View** for this article is available online:
http://emboj.embopress.org

## Acknowledgements

The authors would like to thank Lawrence McIntosh and Mark Okon (University of British Columbia) for acquiring some of the NMR data used in this work. This work was supported by a grant from the Canadian Institutes of Health Research (MOP-14606) and the Canada Research Chairs Program (GSS). This work was supported by Cancer Research UK (A17739), the EMBO Young Investigator Programme and the Medical Research Council (HW), and by a Burroughs Wellcome Fund Collaborative Research Travel Grant (GSS, HW).

## Author contributions

AK crystallized and solved the UblR0RBR and S65DR0RBR structures, performed the complex experiments in Fig 6, analysed data and wrote the manuscript. JDA collected NMR experiments for assignment, conducted NMR and ITC experiments for Ubl, UblS65E and pUbl R0RBR interactions and pUb-modelling experiments (PRE and heteronuclear nOe), analysed data and wrote the manuscript. TECC assigned NMR spectra, conducted NMR and ITC experiments for Ub, UbS65E and pUb R0RBR interactions, and analysed data and wrote the manuscript. RJMT performed ITC experiments and analysis, and expressed and purified multiple proteins. VKC designed and performed ubiquitination assays. RT cloned several constructs. RS performed SEC-MALS analysis in Appendix Fig S6. PM assigned NMR spectra of R0RBR.

AK produced phosphoproteins. DES completed preliminary Ub interaction experiments with R0RBR by NMR and synthesized spin-labelled pUb proteins for pUb–R0RBR interaction experiments. KRB expressed and purified numerous isotopically labelled proteins for NMR experiment. GSS and HW conceived of the study, designed experiments, analysed data and wrote the manuscript.

## Conflict of interest

The authors declare that they have no conflict of interest.

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
