## [Review Process File · The EMBO Journal]

Manuscript EMBO-2015-92337

Disruption of the autoinhibited state primes the E3 ligase parkin for activation and catalysis

Atul Kumar, Jacob D Aguirre, Tara EC Condos, R Julio Martinez-Torres, Viduth K Chaugule, Rachel Toth, Ramasubramanian Sundaramoorthy, Pascal Mercier, Axel Knebel, Donald E Spratt, Kathryn R Barber, Gary S Shaw and Helen Walden

Corresponding author: Helen Walden, MRC Protein Phosphorylation and Ubiquitylation Unit

Review timeline:	Submission date:	18 June 2015
	Editorial Decision:	29 June 2015
	Revision received:	15 July 2015
	Accepted:	23 July 2015

Editor: Karin Dumstrei

Transaction Report:

1st Editorial Decision

29 June 2015

Thank you for submitting your manuscript to The EMBO Journal.

Your manuscript has been previously reviewed at another journal and submitted to The EMBO Journal with referee comments and a point-by-point response. I involved one new expert to act as an arbitrator and to assess the suitability for publication in The EMBO Journal. The referee had access to the previous referees' comments and point-by-point response.

I have now heard back from the referee and as you can see below the referee is supportive of publication here pending some revisions. The referee would like to see the inclusion of additional ubiquitination assays and ITC analysis, which should be fairly straightforward to address. Please also use the current nomenclature for the IRB and RNG2 domains.

Could you let me know the timeframe for doing the revisions as I am keen on getting the revised version back as soon as possible.

When preparing your letter of response to the referees' comments, please bear in mind that this will form part of the Review Process File, and will therefore be available online to the community. For more details on our Transparent Editorial Process, please visit our website: http://emboj.emboress.org/about#Transparent_Process

You can use the link below to upload the revised version

REFeree REPORT

Referee #1:

Parkin is a member of the RBR family of E3 ubiquitin ligases and exists in an autoinhibited state in which access to the E2 binding site is occluded due to intramolecular interactions. Activation of Parkin depends on the activity of PINK1 a kinase which phosphorylates S65 in the Ubl domain of Parkin and S65 of ubiquitin thereby increasing its affinity for Parkin. Both of these activities have been shown to contribute to Parkin activation though the underlying mechanism remains elusive and at present only structures of autoinhibited Parkin are available.

The authors of this manuscript provide the high resolution structure of Parkin in the autoinhibited state (containing an engineered loop between the Ubl and RING0 domains) and the structure of a S65D Parkin mutant as a mimic of the phosphorylated state. The authors complement these crystallographic studies with a range of solution studies to gain insight into the structural changes that occur after phosphorylation of S65 in the Ubl domain and interaction of Parkin with phosphorylated ubiquitin. Based on their data the authors provide a model for the Parkin-pUb complex and explain how these modifications induce conformational changes that promote an active conformation allowing Parkin to ubiquitinate substrate proteins.

This study provides novel and interesting insight into the regulatory mechanisms that underlie Parkin's catalytic activity that includes priming of the pUb binding site by phosphorylation of the Ubl domain. Based on the available data the authors suggest a model how multiple inputs work together to release autoinhibition to promote interaction with the ubiquitin-loaded E2.

Major concerns

- The ubiquitination assays shown in Fig 1B suggest that the wt Parkin with the "long" Ubl-RING0 linker is more active than the crystallized short version. What do the authors think is the reason for this? Please discuss.

It is interesting to see the formation of unanchored ubiquitin chains using fluorescently labelled ubiquitin when in the past people have only reported autoubiquitination of Parkin. Please comment on this new observation and why it might have been missed previously.

- The structures of UblR0RBR and S65D UblR0RBR look very similar. How can the authors be sure that the S65D mutant is a good phosphomimetic? They should compare binding of S65D Ubl to R0RBR with that of pUbl. Are the affinities the same?

Ideally the authors could present ubiquitination assays in which they compare the activity of wt Parkin, UblR0RBR, S65D UblR0RBR and phospho-UblR0RBR.

Is it possible that the results presented in Fig. S8 that show that phosphomimetic Ubl is not able to bind and displace phosphomimetic Ub despite having similar affinities suggest that S65D Ubl might not be an ideal phosphomimetic?

- The same applies to the use of phosphomimetic Ub. This might be acceptable and sufficiently close for enzymatic assays but may not be for a detailed thermodynamic characterization of an interaction. Hence, the authors should repeat the ITC titrations shown in Fig. 3D with the phosphorylated versions of the Ubl domain and ubiquitin if they want to make a thermodynamic argument that goes beyond a simple K_d to ensure they behave in a similar fashion.

Furthermore, the authors should state explicitly what the K_d of the interaction is. It doesn't say anywhere in the text. In fact, the authors should provide a Table listing all the affinities known for the interaction of Parkin and its variants with Ubl/pUbl and Ub/pUb even if they have been measured in other publications such as 29. Without this information it can be difficult to follow the discussion.

- Why did the authors crystallize the S65D Ubl mutant but did the NMR experiments with the S65E mutant and S65E Ub? I appreciate that pParkin didn't crystallize but please explain why different mutants were used in the other experiments and importantly why it shouldn't matter which ones are used.

The authors could show ubiquitination assays comparing the phosphomimetics with the phosphorylated forms and thereby reassure the reader that these differences are not significant.

- What are the concentrations used in the experiments shown in Fig. 3F? How much S65E Ub needs to be added to displace S65E Ubl? Does this make sense in light of their affinities?

- I am not entirely convinced by the argument for a novel ubiquitin-binding site on RING1 after replacement of the Ubl domain. The mutations made show fairly similar effects to those mutations that disrupt the Ubl-RING1 interaction (Fig. 1F). Furthermore, the model of a ubiquitin binding site on RING1 goes against other data (reference 5) that suggested that RBR ligases would not function via a typical RING-type mechanism and do not contact ubiquitin to promote an active conformation (which may not be required for a transthiolation reaction). Is there any evidence that E2~Ub binds tighter than E2? Please discuss and possibly modify in Fig.4 step 4. What is the evidence to support an ubiquitin BRcat/IBR domain interaction?

- The Discussion should be slightly extended to relate the data presented in this manuscript in more detail to previous studies and models for Parkin activation.

Minor concerns

- While I fully agree with the authors that the current nomenclature for the IBR and RING2 domains is inappropriate and should be changed, I am not convinced that this manuscript is the time to do so as everybody else is sticking to the original nomenclature. This is only confusing for the reader. The authors have suggested a novel nomenclature in the past in a review in Biochem J, but this hasn't been generally adopted. Maybe this requires another (general) discussion.

- Fig 1A: Could the authors indicate the regions for which there is no electron density by dashed lines? What region is shown in cyan? I assume the shortened linker? Please specify.

- The symbols above the gel in Fig. 1B are not intuitive. The authors should consider changing them to something more obvious.

- Please add the concentrations at which the HSQC experiments were done to the figure legend of Fig 1E.

- The authors should add an ubiquitination assay using the truncated R0RBR construct to Fig. 1F to show how this compares to the mutants tested.

- The authors need to label the domains in Fig 3A to help the reader orient themselves. Furthermore, it is very difficult to see the different shades of blue mentioned in the figure legend. Maybe use a different colour.

- Why are residues of the basic triad (H302 and R305) that surround the pS65 Ub binding site not undergoing chemical shifts?

- I'm not convinced that Fig. 3E adds anything to the manuscript.

- The authors should describe explicitly in Materials and Methods what residues the crystallized construct contains.

- Please state the protein concentrations used for the ITC titrations and SEC-MALS.

- Have the NMR assignments been deposited? Please add the relevant information.

- There are a few typos in the manuscript. Please check.

1st Revision - authors' response

15 July 2015

Referee #1:

Parkin is a member of the RBR family of E3 ubiquitin ligases and exists in an autoinhibited state in which access to the E2 binding site is occluded due to intramolecular interactions. Activation of

Parkin depends on the activity of PINK1 a kinase which phosphorylates S65 in the Ubl domain of Parkin and S65 of ubiquitin thereby increasing its affinity for Parkin. Both of these activities have been shown to contribute to Parkin activation though the underlying mechanism remains elusive and at present only structures of autoinhibited Parkin are available.

The authors of this manuscript provide the high resolution structure of Parkin in the autoinhibited state (containing an engineered loop between the Ubl and RING0 domains) and the structure of a S65D Parkin mutant as a mimic of the phosphorylated state. The authors complement these crystallographic studies with a range of solution studies to gain insight into the structural changes that occur after phosphorylation of S65 in the Ubl domain and interaction of Parkin with phosphorylated ubiquitin. Based on their data the authors provide a model for the Parkin-pUb complex and explain how these modifications induce conformational changes that promote an active conformation allowing Parkin to ubiquitinate substrate proteins.

This study provides novel and interesting insight into the regulatory mechanisms that underlie Parkin's catalytic activity that includes priming of the pUb binding site by phosphorylation of the Ubl domain. Based on the available data the authors suggest a model how multiple inputs work together to release autoinhibition to promote interaction with the ubiquitin-loaded E2.

Major concerns

- *The ubiquitination assays shown in Fig 1B suggest that the wt Parkin with the "long" Ubl-RING0 linker is more active than the crystallized short version. What do the authors think is the reason for this? Please discuss.*

The role of the linker between the UbID and RING0 is currently unknown. We can speculate that it provides flexibility for the removal of the Ubl domain, and may also provide additional sites of autoubiquitination which may lead to a shortened version looking 'less' active. The length and type of linker in Parkin is the point of most variation, with nematode not having much of a linker at all (see alignment in expanded view fig ev1). However, the presence of the linker precludes high resolution structural analysis. The linker does play a role in activity and to address this we have purified several linker mutants, both with and without the Ubl domain, and present the assays in figure 1A and B. The parkin literature contains many discrepant reports of parkin activity, which we think is likely due to differences in constructs used. Thus we include an activity assay of the proteins used in this study as a reference.

It is interesting to see the formation of unanchored ubiquitin chains using fluorescently labelled ubiquitin when in the past people have only reported autoubiquitination of Parkin. Please comment on this new observation and why it might have been missed previously.

Most parkin assays in the literature, including our previous work, have relied on western blot analysis of assays run on SDS-PAGE, transferred to a membrane, and then blotted with at least 2 antibodies, if not more. We have always found this somewhat unsatisfactory, often the anti-ubiquitin blots don't match the Parkin blots (for example see fig 2G of PMID:23661642) or show Parkin ubiquitination activity that is only detected in the stacking buffer (Figs 2E and 2F of the same paper). These assays are prone to artefacts from incomplete transfers, batch-to-batch variation in antibody detection, antibodies that may have 'gone off', gels that are run further than where ubiquitin would run, membranes that are cut off, different exposure times, and so on. Therefore we wanted to develop an assay that would be reproducible, quantifiable, robust, so we generated a mutant of ubiquitin carrying a cysteine in place of the N-terminus, and attached a fluorophore. Then we run the assays on SDS-PAGE and directly detect in-gel fluorescence. We control with both no E3, and no ATP so it is reasonable to interpret that the bands formed are a mixture of ubiquitin chains on their own and on Parkin and/or E2.

We now include a description of the assay in the materials and methods.

The structures of UbIR0RBR and S65D UbIR0RBR look very similar. How can the authors be sure that the S65D mutant is a good phosphomimetic?

We can't be sure this is a good phosphomimetic. We can however show that S65D-Parkin is activatable in a way similar to phosphoparkin. As is always the case with mimetics, it is not

possible to faithfully completely recapitulate the phosphorylated form, but the assay in Appendix Figure S5 and the table of ITC measurements (Table II) show that the mimetics are partially able to mimic the phosphorylated forms.

They should compare binding of S65D Ubl to R0RBR with that of pUbl. Are the affinities the same?

The affinities of S65D Ubl and S65E Ubl for R0RBR are 5.0 and 5.3 micromolar respectively, while phosphoUbl has a Kd of 26 micromolar. This is a 5-fold difference. The trend is clear – putting a negative charge at position 65 weakens the interaction with R0RBR. We provide affinity measurements for phosphomimetics in Table II.

Ideally the authors could present ubiquitination assays in which they compare the activity of wt Parkin, UblR0RBR, S65D UblR0RBR and phospho-UblR0RBR.

We now include this assay in supplementary figure 5.

Is it possible that the results presented in Fig. S8 that show that phosphomimetic Ubl is not able to bind and displace phosphomimetic Ub despite having similar affinities suggest that S65D Ubl might not be an ideal phosphomimetic?

We think we may not have explained this assay clearly enough. The Ubl domain and the S65E Ubl domain (or S65D – see new Table II) bind to R0RBR with roughly 3-5 micromolar Kd. phosphoUbl binds with 26 micromolar Kd. S65E ubiquitin displaces S65E-Ubl, even though S65E-Ubl actually binds *tighter* than phosphoUbl, so the inability of the phosphomimetic Ubl to displace phosphomimetic ubiquitin suggests that phosphomimetic ubiquitin stays where it is even in the presence of a higher affinity interaction, albeit at a different site. The interpretation of this experiment is that the binding of phosphoubiquitin *displaces* the ubl domain, not replaces it.

The same applies to the use of phosphomimetic Ub. This might be acceptable and sufficiently close for enzymatic assays but may not be for a detailed thermodynamic characterization of an interaction. Hence, the authors should repeat the ITC titrations shown in Fig. 3D with the phosphorylated versions of the Ubl domain and ubiquitin if they want to make a thermodynamic argument that goes beyond a simple Kd to ensure they behave in a similar fashion.

We very much want to understand how the Ubl and ubiquitin bind, in a way that goes beyond simple Kds, hence why we have used ITC to assay the interactions. We have now replaced the ITC titrations previously shown in Figure 3D with the phosphorylated Ubl and ubiquitin, now Figure 4 and described the thermodynamics for the phosphorylated proteins in the Figure 4D and the text. The enthalpies and entropic changes for the phosphoproteins parallel those seen with the mimetics indicating the binding modes are similar.

Furthermore, the authors should state explicitly what the Kd of the interaction is. It doesn't say anywhere in the text. In fact, the authors should provide a Table listing all the affinities known for the interaction of Parkin and its variants with Ubl/pUbl and Ub/pUb even if they have been measured in other publications such as 29. Without this information it can be difficult to follow the discussion.

We now include these data in Table II. We provide Kds, enthalpy and entropy measurements for R0RBR vs Ubl, pUbl, S65DUbl, S65EUbl; and R0RBR vs Ub, pUb, S65DUb, and S65EUb. These data show that S65D or E versions behave comparably. We also include in Table II measurements of Parkin with pUb, and phosphoParkin with pUb, as well as those from the literature.

- *Why did the authors crystallize the S65D Ubl mutant but did the NMR experiments with the S65E mutant and S65E Ub? I appreciate that pParkin didn't crystallize but please explain why different mutants were used in the other experiments and importantly why it shouldn't matter which ones are used.*

We used a variety of mutants because there are different ones used in the literature. Also, as the reviewer pointed out earlier, it can be difficult to predict which, if either, negatively charged amino acid will make a good mimic. The S65D version of UblR0RBR crystallised.

Then a recent paper shows the structure of phosphoubiquitin that it is functionally comparable to S65E ubiquitin (PMID: 25527291). Therefore we decided our NMR analysis, which relies on pure, homogeneous samples, was best done in the context of the E mutations, and given the similarity between the Ubl and Ub moieties, we wanted to keep the NMR consistent. PhosphoParkin doesn't crystallise, yet. We are still trying.

The authors could show ubiquitination assays comparing the phosphomimetics with the phosphorylated forms and thereby reassure the reader that these differences are not significant. We include this assay as Supplementary Figure 5.

- What are the concentrations used in the experiments shown in Fig. 3F? How much S65E Ub needs to be added to displace S65E Ubl? Does this make sense in light of their affinities?

We have added the concentrations to the experiments. The experiment described had 83% bound S65E Ubl to R0RBR based on the Kd. The NMR spectra were consistent with this. Upon addition of an equimolar amount of S65E Ub the spectra reverted back to that of "free S65E Ubl". This is consistent with a similar Kd but different binding site for the S65E Ub. If S65E Ub occupied the same site one would expect about 45% of the S65E Ubl to remain bound and this was clearly not observed. The figure legend and text in the paper were modified to better describe this experiment.

I am not entirely convinced by the argument for a novel ubiquitin-binding site on RING1 after replacement of the Ubl domain. The mutations made show fairly similar effects to those mutations that disrupt the Ubl-RING1 interaction (Fig. 1F).

Furthermore, the model of a ubiquitin binding site on RING1 goes against other data (reference 5) that suggested that RBR ligases would not function via a typical RING-type mechanism and do not contact ubiquitin to promote an active conformation (which may not be required for a transthiolation reaction). Is there any evidence that E2~Ub binds tighter than E2? Please discuss and possibly modify in Fig.4 step 4.

What is the evidence to support an ubiquitin BRcat/IBR domain interaction?

We appreciate that this is a novel suggestion. Single point mutations in the helix 1 of RING1 that are binding the Ubl domain are quite similar to the Ub side, we agree, although not in the context of Miro1 ubiquitination. However, D270R is completely dead (new Figure 6C). In addition, a triple mutant incorporating all the three that are still active in single context is also dead. The reviewer's question as to whether there is evidence that E2~Ub binds tighter than E2 alone we have addressed with 2 new experimental setups. Firstly, we have assayed complex formation by size-exclusion chromatography. pParkin plus pUb shifts relative to pParkin to form a complex (Figure 6D). Addition of UbcH7 does not result in a further shift. Addition of UbcH7~E2 (isopeptide linked through lysine in lieu of catalytic cysteine) does shift, and the gel of each peak shows clearly the 3 protein complex with pParkin/pUb/UbcH7~Ub. No such complex is formed when the UbcH7 is not charged. We then repeated this experiment in the context of a helix 1 RING1 mutant that should be open but unable to bind ubiquitin if our hypothesis is correct. The phosphorylated triple mutant Parkin still forms a complex with pUb, but is now no longer able to form a complex with UbcH7 charged with ubiquitin. Finally we measure the interaction between Parkin complexed with phosphoubiquitin and E2, or E2~Ub, both in the wild-type and triple mutant forms. In addition we have measured the interaction of UbcH7 or UbcH7~Ub with the phosphoparkin/phosphoubiquitin complex by ITC (Figure 6E). We find that the Kd for UbcH7 alone is ~20 micromolar, while the Kd for UbcH7~Ub is ~900 nanomolar supporting a Ub binding site as part of the E2~Ub conjugate.

We have also previously shown that Parkin interacts non-covalently with ubiquitin (Chaugule et al., 2011). This finding was picked up on and developed in a paper from Tony Hunter's group (PMID:23670163). We do not think this necessarily extends to other RBR ligases, nor do we suggest that, especially considering that most RBR E3 ligases do not contain Ubl domains. But we do think a reasonable interpretation of our data is that the displacement of the Ubl domain allows the E2~Ub to bind more effectively, both via the E2, and the ubiquitin moieties. We also think this explains the apparent promiscuity Parkin exhibits towards E2s – if a broad spectrum is needed, the common denominator is ubiquitin.

Finally we do not mean to suggest an interaction between ubiquitin and the IBR. We have modified the model to remove this interpretation.

- The Discussion should be slightly extended to relate the data presented in this manuscript in more detail to previous studies and models for Parkin activation.

We have extended the discussion to include details of other studies for Parkin activation.

Minor concerns

- While I fully agree with the authors that the current nomenclature for the IBR and RING2 domains is inappropriate and should be changed, I am not convinced that this manuscript is the time to do so as everybody else is sticking to the original nomenclature. This is only confusing for the reader. The authors have suggested a novel nomenclature in the past in a review in Biochem J, but this hasn't been generally adopted. Maybe this requires another (general) discussion.

We agree with the reviewer that the IBR RING2 names are wrong not only for parkin but for all RBR E3 ligases and we don't feel entirely comfortable with propagating such an inaccurate nomenclature. We have seen our proposed naming in the review from Riley and Olzmann in PloS Genetics this year. Our Biochemical Journal review has been downloaded over 3000 times in just over a year, and we think it is likely the more accurate names will be picked up. At the request of the editor we have reverted to the IBR and RING2 nomenclature. However, given the inaccuracy of this nomenclature it would be equally appropriate for other structural papers to use the new conventions.

- Fig 1A: Could the authors indicate the regions for which there is no electron density by dashed lines? What region is shown in cyan? I assume the shortened linker? Please specify.

Cyan is simply the colour of the non-domain parts of the molecule. We have clarified this in the legend.

- The symbols above the gel in Fig. 1B are not intuitive. The authors should consider changing them to something more obvious.

We have changed the clock symbols to minutes.

- Please add the concentrations at which the HSQC experiments were done to the figure legend of Fig 1E.

These have been added to Figure 1 and all other Figures with NMR data.

- The authors should add an ubiquitination assay using the truncated R0RBR construct to Fig. 1F to show how this compares to the mutants tested.

We have now included this in figure 1A/B.

- The authors need to label the domains in Fig 3A to help the reader orient themselves. Furthermore, it is very difficult to see the different shades of blue mentioned in the figure legend. Maybe use a different colour.

We agree with the reviewer. The figure has been re-done using a surface with a magenta gradient that makes the regions of maximum chemical shift change more obvious.

- Why are residues of the basic triad (H302 and R305) that surround the pS65 Ub binding site not undergoing chemical shifts?

The NMR assignment of R0RBR is only partially complete. We have assigned K151 (RING0) and this residue in the basic triad undergoes a large shift (also G152). As the reviewer undoubtedly appreciates, the assignment of a 320-residue protein is challenging. We have resorted to specifically labeling the His residues in an attempt to help complete the

assignments but currently do not have H302 and R305 assigned.

- I'm not convinced that Fig. 3E adds anything to the manuscript.

We have removed this figure.

- The authors should describe explicitly in Materials and Methods what residues the crystallized construct contains.

We have explicitly described the construct.

- Please state the protein concentrations used for the ITC titrations and SEC-MALS.

All concentrations have been added in the Materials and Methods.

- Have the NMR assignments been deposited? Please add the relevant information.

As requested we have deposited several chemical shift data sets to the BMRB and added accession numbers to the manuscript.

- There are a few typos in the manuscript. Please check.

We have tried to find them all.

2nd Editorial Decision

22 July 2015

Thank you for submitting your revised manuscript to The EMBO Journal. Your manuscript has now been re-reviewed by the referee. I am happy to say that the referee appreciates the introduced changes. I am therefore very pleased to accept the manuscript for publication here.

REFeree REPORT

Referee #1:

The authors have fully addressed all my comments and the inclusion of new experiments and changes to the text have substantially improved the manuscript. This is now a very nice story that will be of great interest to a wide readership.